# Control of a programmed cell death pathway in *Pseudomonas aeruginosa* by an antiterminator

Jennifer M. Peña[1], Samantha M. Prezioso[1,4], Kirsty A. McFarland[1,4], Tracy K. Kambara[1], Kathryn M. Ramsey [1,2], Padraig Deighan[3] & Simon L. Dove [1✉]

In *Pseudomonas aeruginosa* the *alp* system encodes a programmed cell death pathway that is switched on in a subset of cells in response to DNA damage and is linked to the virulence of the organism. Here we show that the central regulator of this pathway, AlpA, exerts its effects by acting as an antiterminator rather than a transcription activator. In particular, we present evidence that AlpA positively regulates the *alpBCDE* cell lysis genes, as well as genes in a second newly identified target locus, by recognizing specific DNA sites within the promoter, then binding RNA polymerase directly and allowing it to bypass intrinsic terminators positioned downstream. AlpA thus functions in a mechanistically unusual manner to control the expression of virulence genes in this opportunistic pathogen.

[1] Division of Infectious Diseases, Boston Children's Hospital, Harvard Medical School, Boston, MA, USA. [2] Departments of Cell and Molecular Biology and Biomedical and Pharmaceutical Sciences, University of Rhode Island, Kingston, RI, USA. [3] Department of Biology, Emmanuel College, Boston, MA, USA. [4] These authors contributed equally: Samantha M. Prezioso, Kirsty A. McFarland. ✉email: simon.dove@childrens.harvard.edu

*P*seudomonas aeruginosa is an opportunistic pathogen of humans and is a leading cause of nosocomial infections and chronic lung infections of cystic fibrosis patients[1–3]. The ability of *P. aeruginosa* to adapt to and thrive within a variety of environmental niches, including those of the host, is thought in part to be due to the large number of transcription regulators the organism encodes[3,4]. Although the majority of transcription regulators that have been studied in *P. aeruginosa*, as well as other bacteria, exert their regulatory effects at the level of transcription initiation, essentially any step in the transcription cycle is a potential target for regulation, including transcription termination. Indeed, antiterminators comprise one important class of positive regulator that acts at a post-initiation step. Members of this class broadly function by inhibiting transcription termination, thus allowing for transcription to continue into genes that are positioned downstream of termination sites[5]. Some antiterminators act at a single specific termination site, whereas so-called processive antiterminators modify RNA polymerase (RNAP), making it resistant to termination signals and allowing it to bypass many termination sites[6,7]. Thus far, relatively few processive antiterminators have been identified[7,8], and only the NusG-related RfaH protein and its orthologs have been implicated in the control of virulence gene expression in any bacterium[9,10]. *P. aeruginosa* does not appear to encode an ortholog of RfaH[11] and no processive antiterminator has previously been reported in this organism.

Contributing to the virulence of *P. aeruginosa* is a programmed cell death (PCD) pathway that is triggered in a stochastic fashion in response to DNA damage[12]. This PCD pathway is encoded by the genes *alpRABCDE*. AlpR is a repressor that undergoes autocleavage in response to DNA damage resulting in derepression of *alpA*[12]. AlpA then goes on to positively regulate the expression of the *alpBCDE* genes resulting in cell lysis. How AlpA regulates the *alpBCDE* cell lysis genes was not known, nor was it known whether AlpA controlled other genes in addition to *alpBCDE*.

Here we present evidence that AlpA positively regulates target gene expression by recognizing specific sites on the DNA, then loading onto RNAP and allowing it to bypass intrinsic termination sites positioned downstream of target promoters, thus functioning as a processive antiterminator. Additionally, we present evidence that AlpA functions in the same manner to positively regulate the expression of genes in a second putative operon comprising genes *PA0807–PA0829*. This second AlpA-regulated locus does not appear to contribute to cell lysis; however, genes within this locus are known to be important for the virulence of *P. aeruginosa*[13,14]. Lastly, we provide evidence that the activity of AlpA is stimulated by the small molecule guanosine tetraphosphate (ppGpp), which is better known for its control of the stringent response in bacteria. Our findings suggest that when produced in response to DNA damage AlpA functions as a processive antiterminator to regulate virulence gene expression in a manner that may be modulated by the intracellular concentration of ppGpp.

## Results

**AlpA positively regulates the expression of genes in two distinct operons.** AlpA is a positive regulator of the *alpBCDE* genes which encode a self-lysis cassette[12,15]. To determine whether AlpA controls the expression of other genes in addition to *alpBCDE*, we treated cells of *P. aeruginosa* strain PAO1, cells of an *alpA* deletion mutant (PAO1 Δ*alpA*), and cells of an *alpA* (stop) mutant (that contains a stop codon early on in the *alpA* gene; PAO1 *alpA*(stop)) with ciprofloxacin, an antibiotic known to cause DNA damage and induce the expression of *alpA*[12]. Comparison of the transcriptomes of cells of the three different strains by RNA-Seq indicated that in addition to positively regulating the expression of the *alpBCDE* genes, AlpA positively regulates the expression of genes present in a separate region of the chromosome *PA0807–PA0829* (Fig. 1a, b) (Supplementary Data 1 and 2). Specifically, we found that AlpA positively regulates the expression of all of the genes on the positive strand between *PA0807* and *PA0829* (i.e. *PA0807* (*ampDh3*), *PA0808*, *PA0815*, *PA0817*, *PA0819*, *PA0820*, *PA0828*, and *PA0829*) (Fig. 1a, b), a finding we confirmed using quantitative reverse transcriptase PCR (qRT-PCR) (Supplementary Fig. 1). We note that many of the genes that appear to be negatively regulated by AlpA when comparing the transcriptomes of Δ*alpA* mutant cells to those of wild-type (WT) PAO1 (Fig. 1a and Supplementary Data 1) are controlled by a regulator we identified previously called BexR, which together with its regulon is expressed in a bistable fashion[16]. We suspect that the BexR-regulated genes only

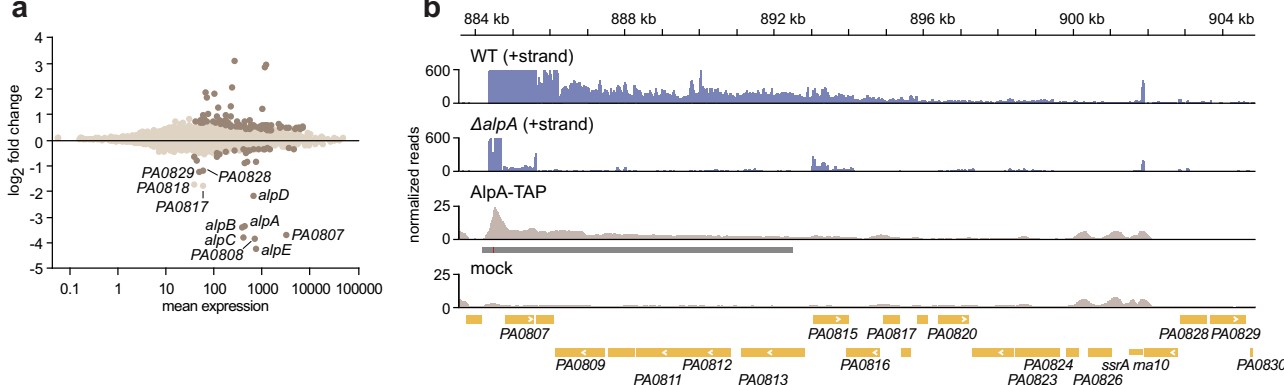

**Fig. 1 AlpA positively regulates the expression of genes in two operons in *P. aeruginosa*. a** Plot of RNA-Seq data comparing gene expression in PAO1 Δ*alpA* to PAO1 WT. Cells were treated with 1 µg/mL ciprofloxacin for 2 h to induce DNA damage. Brown dots are statistically significantly differentially regulated genes, beige dots are genes that are not significantly differentially regulated. Genes shown to be most highly positively regulated by AlpA are labeled. **b** RNA-Seq data from PAO1 WT and PAO1 Δ*alpA* mutant cells (upper panels) treated with ciprofloxacin, and ChIP-Seq data from cells synthesizing AlpA-TAP (lower panels) depicting reads over the *PA0807–PA0830* region. For RNA-Seq, only reads mapping to the plus strand (indicated +strand) are shown (in blue). ChIP-Seq of AlpA-TAP shows AlpA associates with the *PA0807* promoter region and operon. Significantly enriched peaks are indicated by a dark gray box below the read density plot (in beige), red line within this dark gray box indicates site of maximum enrichment. Control (indicated mock) is the mock IP control performed with PAO1 cells that synthesize AlpA without a TAP-tag. Genes are depicted in yellow at the bottom of the panel. Genomic location in kb is provided at the top of the panel.

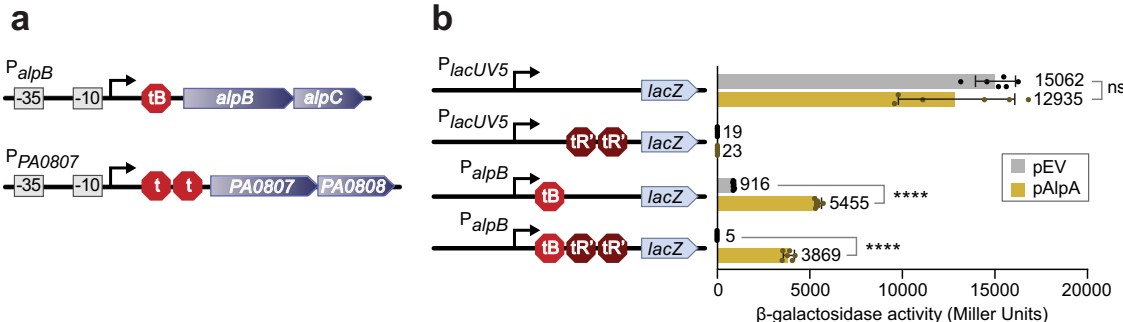

**Fig. 2 AlpA allows RNAP to bypass multiple intrinsic termination sequences located downstream of the *alpB* promoter. a** Diagram of promoter regions of the two operons AlpA positively regulates. There is one intrinsic terminator, tB, predicted upstream of the *alpB* ORF and two putative intrinsic terminators (depicted as red hexagons) positioned upstream of the *PA0807* ORF. **b** β-galactosidase activity (in Miller Units) of the indicated reporter constructs in *E. coli*. Promoters were either *lacUV5* or *alpB* and terminators are tB (red hexagon) or tR′ (brown hexagon). Cells contained plasmid pAlpA (gold) or plasmid pPSV38 (the empty vector control indicated pEV; gray). Values and error bars reflect mean ± SD of *n* = 3 biological replicates in technical duplicate; two-tailed, unpaired, unequal variance *t*-tests were used to calculate *p*-values between indicated samples. P$_{alpB}$ tB reporter with pEV vs. pAlpA: *p* = 4.8 × 10$^{-9}$. P$_{alpB}$ tB tR′tR′ reporter with pEV vs. pAlpA: *p* = 4.1 × 10$^{-7}$. *p*-Values indicated by the following symbols: >0.05 = ns, ≤0.0001 = ****. Source data are provided as a Source Data file.

appear to be upregulated in the Δ*alpA* mutant (Supplementary Data 1) because by chance the colonies used to inoculate these cultures originated from a cell in which *bexR* is expressed (leading to elevated expression of the entire BexR regulon) whereas the colonies used to inoculate the cultures of the WT control originated from a cell in which *bexR* is not expressed. Consistent with this notion, no BexR-regulated genes are identified as being subject to control by AlpA when the gene expression profiles of the *alpA*(stop) mutant cells are compared to those of WT (Supplementary Data 2). Taken together, our RNA-Seq studies indicate that in response to DNA damage, AlpA acts to positively regulate the *alpBCDE* genes as well as those genes encoded on the positive strand from *PA0807–PA0829*.

To determine which genes were controlled directly by AlpA, we performed chromatin immunoprecipitation together with high-throughput DNA sequencing (ChIP-Seq) with cells that ectopically synthesize AlpA with a C-terminal tandem affinity purification (TAP)-tag (AlpA-TAP). These ChIP-Seq analyses revealed that AlpA associates with six regions of the PAO1 chromosome, with the region most enriched for AlpA beginning 728 bp upstream of *PA0807*, in the promoter region, and extending 12.2 kb downstream (Fig. 1b) (Supplementary Table 1). These findings suggest that AlpA exerts its regulatory effects on *PA0807–PA0829* directly. AlpA did not detectably associate with the *alpB* promoter region as assessed by ChIP-Seq (Supplementary Table 1). AlpR binding to the *alpB* promoter region during our ChIP-Seq studies may have prevented the binding of AlpA to this region[12], explaining our inability to detect it at this location. Consistent with this possibility, ChIP followed by quantitative PCR (qPCR) indicated that AlpA-TAP associates with the *alpB* promoter in cells treated with ciprofloxacin—conditions under which AlpR is inactivated through autocleavage[12] (Supplementary Fig. 2). In further support of the notion that AlpA controls the expression of *alpBCDE* directly, we have shown previously that AlpA can positively regulate the expression of an *alpB* promoter-*lacZ* fusion in *Escherichia coli*[12].

**AlpA allows RNAP to bypass termination sequences positioned downstream of target promoters.** The ChIP-Seq peak for AlpA at the *PA0807–PA0829* locus is unusual as one shoulder of the peak extends 3′ of the promoter for more than 12 kb (Fig. 1b). The association of AlpA with the genomic region downstream of the putative *PA0807* promoter led us to hypothesize AlpA was associated with RNAP as it transcribes this region. Indeed, the pattern of enrichment we observe for AlpA by ChIP-Seq at the *PA0807* locus is reminiscent of that observed by ChIP for the Q protein of bacteriophage λ (λQ), a transcription antiterminator that gains access to the transcription elongation complex on the phage late gene operon after binding a specific DNA sequence embedded in the late gene promoter[17,18]. Although AlpA bears no sequence homology to λQ or any other transcription antiterminator, the possibility that AlpA exerts its regulatory effects by acting as an antiterminator is supported by the observation that both the *alpBCDE* operon and the putative *PA0807–PA0829* operon contain predicted intrinsic terminators, including ones located immediately upstream of the *alpB* and *PA0807* genes (as depicted in Fig. 2a).

To determine where the *alpB* promoter was positioned in relation to the predicted intrinsic transcription terminator located upstream of *alpB*, we sought to define the *alpB* promoter sequence. To do this, we first identified the putative *alpB* transcription start site (TSS) using RNA-Seq data obtained from PAO1 cells grown in LB to mid-log. Using a reporter construct containing DNA from −153 to +117 relative to the *alpB* TSS, we introduced mutations into the putative −10 and −35 elements of the promoter we had predicted, that would be expected to either abolish, or severely reduce activity. The results depicted in Supplementary Fig. 3 establish the importance of the predicted −10 and −35 elements for *alpB* promoter activity and indicate that the predicted intrinsic transcription terminator present upstream of *alpB* (referred to henceforth as tB) is positioned beginning 54 nucleotides downstream from the TSS. To test if AlpA positively regulated target genes by functioning as a processive antiterminator, we created an *E. coli* reporter strain in which two heterologous intrinsic transcription terminators (two copies of tR′ from bacteriophage λ) were positioned between the *alpB* promoter and a *lacZ* reporter gene on an F′ episome. As controls we employed an analogous *E. coli* reporter strain in which tandem tR′ terminators were positioned between the *lacUV5* promoter and *lacZ*, as well as a reporter strain in which just the *lacUV5* promoter was positioned upstream of *lacZ*. Expression of *alpA* resulted in a greater than 300-fold increase in *lacZ* expression only in those cells that contained the tandem tR′ terminators positioned between the *alpB* promoter and *lacZ* (Fig. 2b). Thus, AlpA allows RNAP to bypass heterologous transcription terminators positioned downstream of the *alpB* promoter but not the *lacUV5* promoter. Furthermore, consistent with the idea that there is a transcription terminator (tB)

positioned between the *alpB* promoter and the *alpB* gene, removal of tB resulted in a greater than 28-fold increase in expression of a *lacZ* reporter gene positioned downstream of the *alpB* promoter and 5′ untranslated region containing tB (Supplementary Fig. 4a). Moreover, expression of *alpA* resulted in a greater than sevenfold increase in expression of a reporter construct that contained both the *alpB* promoter and tB positioned upstream of *lacZ* (Fig. 2b). Taken together, these findings suggest that AlpA positively regulates the expression of the *alpBCDE* operon and the *PA0807–0829* locus by functioning as a transcription antiterminator. Moreover, because AlpA enables RNAP to bypass heterologous transcription terminators such as tR′, AlpA's ability to function as an antiterminator does not appear to be dependent upon the specific sequence of the terminator itself.

**A putative AlpA binding element is located between the −10 and −35 elements of target promoters.** The Phyre2 structural prediction algorithm indicates that AlpA may contain a winged helix DNA-binding motif (Supplementary Fig. 5), suggesting that the ability of AlpA to bind the DNA may be important for its activity. In support of the idea that AlpA may be a DNA-binding protein, our ChIP analyses suggest that AlpA associates with the *alpB* and *PA0807* promoter regions (Fig. 1b and Supplementary Fig. 2). To begin to define the potential sequence requirements for AlpA binding, we first sought to identify a minimal region of the *alpB* promoter that was responsive to AlpA. To do this, we made a series of *alpB* promoter-*lacZ* fusions that contained different amounts of DNA flanking the *alpB* TSS followed by tandem tR′ terminators and then *lacZ* in *E. coli*. The results depicted in Supplementary Fig. 6a establish that the region of the *alpB* promoter that extends from −37 to +5 is sufficient to confer control by AlpA. (Note that the different reporters analyzed in Supplementary Fig. 6a contain different 5′-untranslated regions upstream of *lacZ* that could account for the differences in the absolute β-galactosidase activities obtained using these different reporters.)

Next, we identified the putative TSS for the *PA0807* promoter as well as the −10 and −35 elements of the *PA0807* promoter in much the same manner as we identified those for the *alpB* promoter (Supplementary Fig. 3). This placed one predicted transcription terminator beginning 26 bp downstream of the *PA0807* TSS and another beginning 247 bp downstream[19], both of which are located upstream of the *PA0807* ORF (see depiction in Fig. 2a). Consistent with the idea that this element contains at least one terminator, removal of DNA containing both of the putative terminators resulted in an approximately sixfold increase in expression of a reporter construct harboring the *PA0807* promoter in cells of PAO1 (Supplementary Fig. 4b). Furthermore, consistent with there being at least one transcription terminator positioned 26 bp downstream of the *PA0807* promoter, analysis of a series of *PA0807* promoter-*lacZ* reporter constructs in *P. aeruginosa*, to which no additional terminator sequence had been added, identified DNA from −56 to +91 as responsive to AlpA in *P. aeruginosa* (Supplementary Fig. 6b). Finally, we found that when placed upstream of tandem tR′ terminators followed by *lacZ*, the region of the *PA0807* promoter from −100 to +5 was responsive to AlpA (Supplementary Fig. 6c). Taken together, our findings suggest that the minimal region of the *PA0807* promoter that is sufficient to confer control by AlpA, when positioned upstream of a transcription terminator, is −56 to +5.

We next compared the minimal portion of the *alpB* promoter that was responsive to AlpA to the corresponding region of the *PA0807* promoter. This revealed sequences positioned between the −35 and −10 elements of the *alpB* and *PA0807* promoters that were common to both and thus might be important for AlpA

binding (Fig. 3a). To determine whether these sequences were critical for AlpA-dependent control of the *alpB* promoter, point mutations were introduced at positions −25 through −20 of an *alpB* promoter-*lacZ* fusion (Fig. 3a). The results depicted in Fig. 3b indicate that the reporter construct containing these six mutations in the *alpB* promoter (denoted P$_{alpB}$ABE6) was considerably less responsive to AlpA than the corresponding WT construct (denoted P$_{alpB}$). Indeed, we found that as few as four point mutations at positions −25 through −22 were sufficient to severely abrogate AlpA-mediated control (Supplementary Fig. 6d). These findings establish that residues positioned between the −10 and −35 elements of the *alpB* promoter are important for AlpA to mediate its regulatory effect.

The sequence we identified in the *alpB* promoter as being critical for AlpA-dependent control is also present at an identical position within the *PA0807* promoter (Fig. 3a). Introduction of mutations at positions −25 through −22 of a *PA0807* promoter-*lacZ* fusion (Fig. 3a) rendered expression of *lacZ* considerably less responsive to AlpA (Supplementary Fig. 6d). Moreover, combining mutations at positions −25 through −20 and position −14 (denoted P$_{PA0807}$ABE7), a position that may be involved in contacting AlpA (at least at the *PA0807* promoter), resulted in a mutant promoter that was even less responsive to AlpA (Fig. 3c and Supplementary Fig. 6e). Taken together, these findings demonstrate that conserved sequences positioned between the −10 and −35 elements of the *alpB* and *PA0807* promoters are critical for AlpA-dependent control.

To determine whether the conserved sequences within the *alpB* and *PA0807* promoters were important for AlpA's association with the DNA, we performed ChIP with ectopically produced AlpA-TAP in WT cells of *P. aeruginosa* as well as in cells containing either the P$_{alpB}$ABE6 or P$_{PA0807}$ABE7 mutations at their respective native chromosomal loci. The results of qPCR analyses following ChIP with AlpA-TAP show that the P$_{alpB}$ABE6 or P$_{PA0807}$ABE7 mutations reduce the association of AlpA with the *alpB* and *PA0807* promoter regions, respectively (Fig. 3d, e). Taken altogether, our findings support the idea that AlpA may be a DNA-binding protein that recognizes a specific sequence element we refer to as the AlpA binding element (ABE), positioned between the −10 and −35 elements of the *alpB* and *PA0807* promoters, and that this putative ABE is important for the positive regulation of target genes by AlpA.

**The putative ABE in the *alpB* promoter is important for AlpA-dependent cell lysis.** To test the physiological importance of the putative ABE in the *alpB* promoter, and to test whether AlpA-dependent control of the *PA0807–PA0829* operon contributes to cell lysis, we compared the effects of inducing the expression of *alpA* in WT cells of *P. aeruginosa*, and in cells that contained either the P$_{alpB}$ABE6 or P$_{PA0807}$ABE7 mutations at the native chromosomal loci. To do this, we employed a dominant negative mutant strategy we had used previously. In particular, we ectopically synthesized the C-terminal domain of AlpR (AlpR-CTD) in cells of each of these strains; production of the AlpR-CTD sequesters AlpR into inactive heterodimers resulting in the derepression of *alpA*[12]. In WT cells, as well as in cells harboring the P$_{PA0807}$ABE7 mutation, the isopropyl-β-D-thiogalactopyranoside (IPTG)-inducible synthesis of the AlpR-CTD resulted in a dramatic reduction in cell growth or viability (Fig. 3f). However, in cells harboring the P$_{alpB}$ABE6 mutation, or in cells in which the *alpBCDE* genes were deleted, ectopic synthesis of the AlpR-CTD had little to no effect on growth or cell viability (Fig. 3f). Consistent with these findings, time-lapse microscopy indicated that ectopic synthesis of the AlpR-CTD resulted in the lysis of otherwise WT cells of *P. aeruginosa* that synthesized mcherry, but

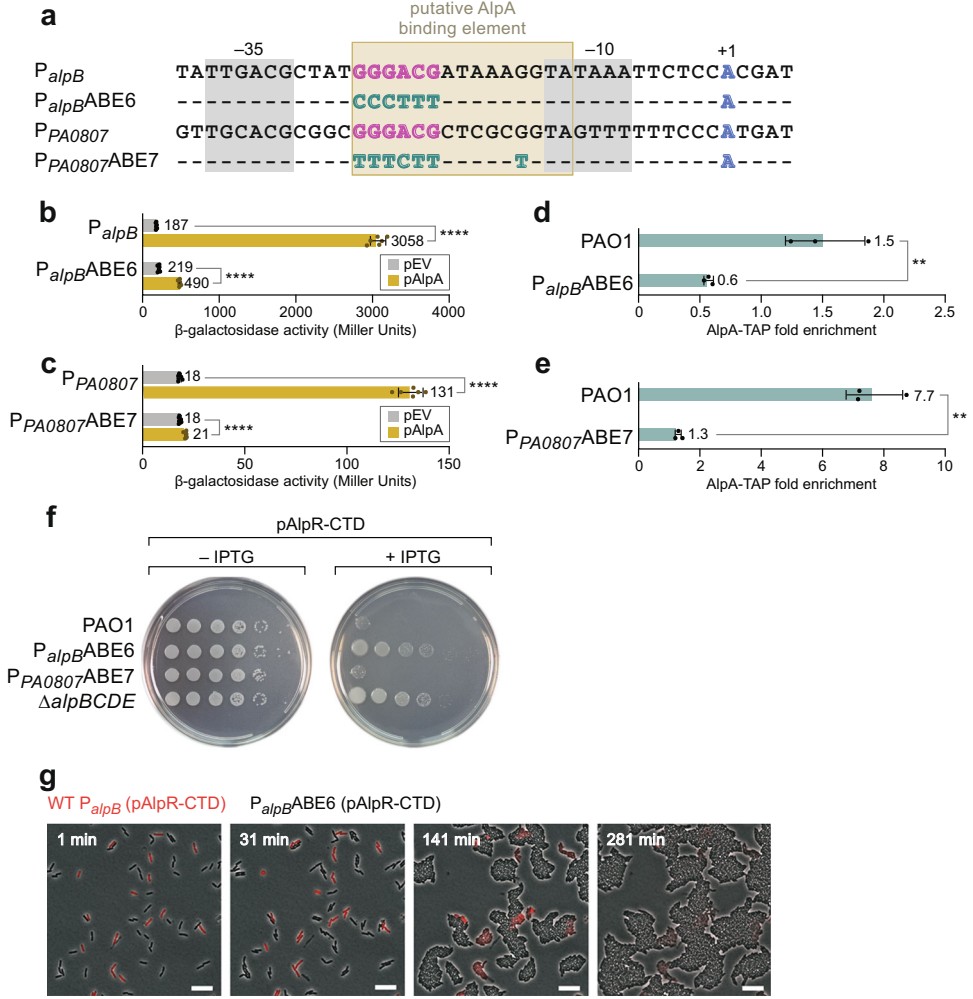

**Fig. 3 A putative AlpA binding element is located between the −35 and −10 elements of target promoters. a** Sequence of *alpB* and *PA0807* promoters and point mutations made in ABE6 and ABE7 promoter mutants. Mutations colored in green, −35 and −10 elements highlighted in gray, TSS in blue. Putative AlpA binding motif is GGGACGN(5)GGTA. **b** Effect of ABE6 mutation on AlpA-dependent control of a P*alpB* tB *lacZ* transcriptional reporter in PAO1. **c** Effect of ABE7 mutation on AlpA-dependent control of a P*PA0807* *lacZ* transcriptional reporter in PAO1. **b, c** Cells of the indicated reporters containing plasmids pAlpA or pPSV38 (the empty vector control, indicated pEV) were assayed for β-galactosidase activity, which is shown in Miller Units. Values and error bars reflect mean ± SD of $n = 3$ biological replicates in technical duplicate. Two-tailed, unpaired, unequal variance *t*-tests were used to calculate *p*-values between indicated samples. P*alpB* with pEV vs. pAlpA: $p = 1.1 \times 10^{-8}$. P*alpB*ABE6 with pEV vs. pAlpA: $p = 8.5 \times 10^{-12}$. P*PA0807* with pEV vs. pAlpA: $p = 5 \times 10^{-8}$. P*PA0807*ABE7 with pEV vs. pAlpA: $p = 6.6 \times 10^{-6}$; $p \leq 0.0001 = ****$. **d, e** Effect of ABE6 and ABE7 mutations on association of AlpA with the respective *alpB* and *PA0807* promoter regions as determined by ChIP-qPCR. Values and error bars reflect mean ± SD of $n = 3$ biological replicates. Two-tailed, unpaired, unequal variance *t*-tests were used to calculate *p*-values between indicated samples. PAO1 (WT P*alpB*) vs. P*alpB*ABE6: $p = 0.01$. PAO1 (WT P*PA0807*) vs. P*PA0807*ABE7: $p = 0.0059$. *p*-Values indicated by the following symbols: $\leq 0.01 = **$. **f** Ectopic synthesis of the AlpR-CTD results in cell death or inhibition of growth in WT PAO1 and PAO1 harboring the ABE7 mutation in the AlpA binding site of the endogenous *PA0807* promoter (P*PA0807*ABE7) but not in PAO1 with a deletion of *alpBCDE* or in PAO1 harboring the ABE6 mutation in the endogenous *alpB* promoter (P*alpB*ABE6). Plasmid pAlpR-CTD directs the synthesis of the AlpR-CTD in an IPTG-inducible manner. Indicated PAO1 strains were serially diluted tenfold and incubated on media lacking or containing IPTG. **g** Time-lapse microscopy images at the indicated time points of PAO1 mCherry with a WT *alpB* promoter (synthesizing mCherry and shown in red) and PAO1 GFP P*alpB*ABE6 (not synthesizing mCherry, shown in phase contrast) ectopically synthesizing the AlpR-CTD. Scale bar is 10 μm. Experiment was performed independently twice with similar results. Source data are provided as a Source Data file.

not of cells that harbored the P*alpB*ABE6 mutation and did not synthesize mcherry (Fig. 3g and Supplementary Movie 1). These findings demonstrate that the putative ABE within the *alpB* promoter is critical for AlpA-mediated cell lysis and suggest that unlike the *alpBCDE* operon, the AlpA-regulated *PA0807–PA0829* locus does not appreciably contribute to cell lysis.

**AlpA binds the β-flap of RNAP and region 1.1 of σ70.** The association of AlpA with the genomic region downstream of the *PA0807* promoter revealed by our ChIP-Seq analyses is consistent with AlpA associating with RNAP as it transcribes the putative

*PA0807–PA0829* operon (Fig. 1b). Indeed, several of the few antiterminators that have been characterized to date act by contacting both the DNA and RNAP[20–24]. To identify potential contact sites on RNAP for AlpA, we used a bacterial two-hybrid system[25]. In this version of the system, predicted surface-exposed portions of *E. coli* RNAP were fused to the CI protein of bacteriophage λ (λCI), whereas AlpA was fused to the portion of the α subunit of *E. coli* RNAP that spans the N-terminal domain and flexible linker region (Fig. 4a). In the *E. coli* reporter strain used here, any sufficiently strong protein–protein interaction between a DNA-bound λCI fusion protein and an RNAP-incorporated

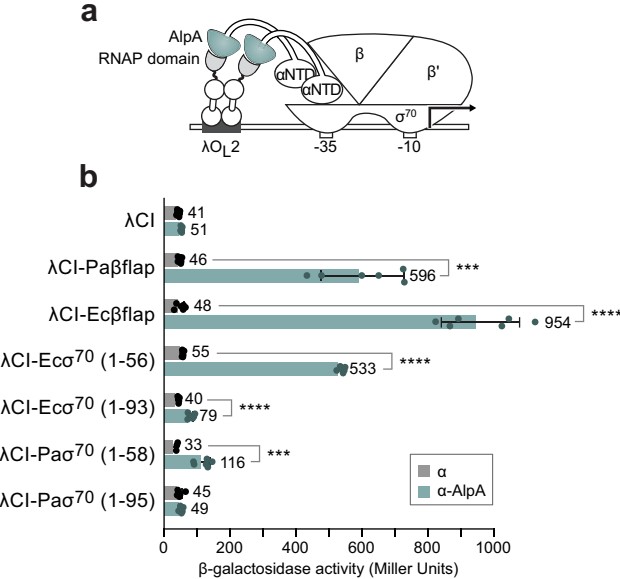

**Fig. 4 AlpA interacts with the β-flap and σ⁷⁰ region 1.1 of RNAP.**
**a** Diagram of bacterial two-hybrid assay used to test whether AlpA interacts with portions of RNAP. Diagram depicts how the interaction between AlpA (green), fused to the α-N-terminal domain and linker (α-NTD), and the RNAP domain (gray), fused to the bacteriophage λCI protein, activates transcription from the test promoter placeO$_L$2-62, which bears the λ operator O$_L$2 centered 62 bp upstream of the start site of the *lac* core promoter driving *lacZ*. **b** Results of β-galactosidase assays performed with cells that contained plasmids directing the synthesis of the indicated proteins under the control of (IPTG)-inducible promoters; cells were grown in the presence of 5 μM IPTG. Values and error bars reflect mean ± SD of $n = 3$ biological replicates in technical duplicate. Two-tailed, unpaired, unequal variance *t*-tests were used to calculate *p*-values between indicated samples. λCI-Paβflap, α vs. α-AlpA: $p = 0.00011$. λCI-Ecβflap, α vs. α-AlpA: $p = 6.5 \times 10^{-6}$. λCI-Ecσ⁷⁰ (1–56), α vs. α-AlpA: $p = 1.4 \times 10^{-10}$. λCI-Ecσ⁷⁰ (1–93), α vs. α-AlpA: $p = 2.3 \times 10^{-5}$. λCI-Paσ⁷⁰ (1–58), α vs. α-AlpA: $p = 0.0002$. *p*-Values indicated by the following symbols: $\leq 0.001 = ***$, $\leq 0.0001 = ****$. Source data are provided as a Source Data file.

α-AlpA fusion would be expected to activate transcription from a test promoter harboring a λ operator positioned at a suitable distance upstream, resulting in a concomitant increase in expression of the linked *lacZ* reporter gene (Fig. 4a). From an initial screen of 59 fragments of *E. coli* RNAP core subunits and σ factors (Supplementary Fig. 7), we detected an interaction between AlpA and the so-called β-flap of *E. coli* RNAP as well as region 1.1 of *E. coli* σ⁷⁰ (refs. [22,26,27]). (Note that we expected AlpA to be able to interact with *E. coli* RNAP because we had already established that AlpA can act as an antiterminator in *E. coli*.) Additional two-hybrid assays indicated that AlpA could detectably interact with the β-flap of *P. aeruginosa* RNAP as well as region 1.1 of *P. aeruginosa* σ⁷⁰ (Fig. 4b). The results of our two-hybrid assays raise the possibility that AlpA may exert its regulatory effects through interaction with the β-flap and/or region 1.1 of σ⁷⁰.

**Ectopic expression of *alpA* is lethal to cells of *P. aeruginosa* even in the absence of the *alpBCDE* cell lysis genes.** We observed that sufficiently high ectopic expression of *alpA* was lethal to cells of *P. aeruginosa* (Fig. 5a). Although we had initially thought the lethal effects of AlpA would be attributable to its effects on expression of the *alpBCDE* cell lysis genes, we unexpectedly found that ectopic expression of *alpA* could be lethal to cells regardless of whether the *alpBCDE* genes were present

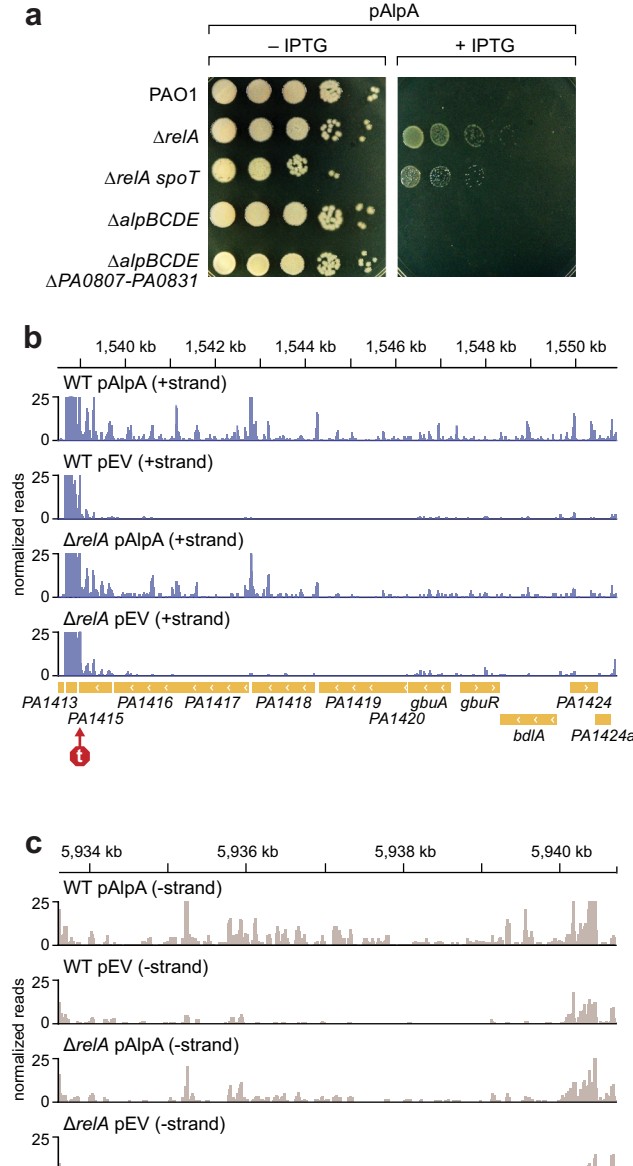

**Fig. 5 Ectopic expression of *alpA* is lethal independent of *alpBCDE*.**
**a** Image shows tenfold spot dilutions of cells of the indicated strains of *P. aeruginosa* ectopically expressing *alpA* from a plasmid (pAlpA) and under the control of an IPTG inducible promoter. PAO1 indicates WT, PAO1 Δ*relA* is indicated Δ*relA*, PAO1 Δ*relA* Δ*spoT* is indicated Δ*relA* Δ*spoT*, PAO1 Δ*alpBCDE* is indicated Δ*alpBCDE*, PAO1 Δ*alpBCDE* Δ*PA0807–PA0831* is indicated Δ*alpBCDE* Δ*PA0807–PA0831*. **b**, **c** Examples in which ectopic expression of *alpA* results in an increase in expression of antisense transcripts. RNA-Seq was performed with PAO1 (WT) and PAO1 Δ*relA* (Δ*relA*) containing pAlpA or pEV (empty vector control plasmid). Transcript abundance shown as peaks. Purple indicates transcripts aligned to the plus strand (+strand). Beige indicates transcripts aligned to the minus strand (−strand). Genes depicted as yellow arrows, arrows pointing to the right are genes encoded on the plus strand, arrows pointing to the left are genes encoded on the minus strand. Images show an increase in transcripts that align to the opposite strand the genes are encoded on (antisense) in cells containing pAlpA. This increase in antisense transcripts corresponds to transcription continuing downstream of predicted intrinsic terminator sequences as indicated with a red arrow.

(Fig. 5a). This indicated that high-level ectopic expression of *alpA* is lethal for reasons other than activating the cell-lysis genes encoded by *alpBCDE*. In addition, we found that ectopic expression of *alpA* was lethal to cells lacking the *alpBCDE* and *PA0807–PA0831* genes (Fig. 5a). In order to determine whether there might be additional genes that AlpA controls when ectopically produced that might result in lethality, we used RNA-Seq. We found that ectopic expression of *alpA* in PAO1 cells resulted in a more than twofold change in expression of 1253 genes or antisense transcripts, 806 of which were upregulated and 447 downregulated (Supplementary Data 3). Strikingly, we noticed that for many genes and antisense transcripts that were positively regulated by AlpA, the regulator appeared to promote the read-through of predicted transcription terminators present at the end, or within, a variety of operons (Fig. 5b, c). The increases in gene expression that are observed specifically when AlpA was ectopically produced might reflect AlpA's association with RNAP off the DNA, promoting the readthrough of transcription terminators which in turn might account for AlpA's toxicity under these conditions in cells lacking *alpBCDE* (Fig. 5a). Furthermore, the negative effects of AlpA on gene expression that are specifically observed upon ectopic synthesis of AlpA might result from AlpA's direct interaction with the DNA or with RNAP, or indirectly through AlpA's positive effects on gene expression (e.g. via overproduction of a negative regulator).

**The alarmone ppGpp may modulate the activity of AlpA.** In an attempt to identify genes that might contribute to the toxic effects of ectopically produced AlpA, we used transposon mutagenesis. In particular, we mutagenized PAO1 with a mariner transposon and then isolated mutants that could tolerate ectopic expression of *alpA*. Mutants were identified with insertions in four different genes: *relA*, *spoT*, *rpoZ*, and *PA4114*. Two of these genes are involved in the synthesis of the alarmone ppGpp; RelA is a ppGpp synthetase whereas SpoT can both synthesize and degrade ppGpp[28,29]. Note that because the *rpoZ* gene is located immediately upstream of *spoT*, we entertained the idea that mutants with insertions in *rpoZ* may have been isolated because the insertions resulted in polar effects on the expression of *spoT*. To test further the involvement of *relA* and *spoT* in mediating sensitivity to AlpA toxicity, mutants with in-frame deletions of either *relA* (PAO1 Δ*relA*) or both *relA* and *spoT* (PAO1 Δ*relA* Δ*spoT*) were constructed. Consistent with what we found with the corresponding transposon insertion mutants, cells of both the Δ*relA* and Δ*relA* Δ*spoT* mutants could better tolerate the ectopic synthesis of AlpA than WT cells by at least three orders of magnitude (Fig. 5a). Moreover, the abundance of ectopically produced epitope-tagged AlpA was only 20% lower in cells of the Δ*relA* mutant and 30% lower in cells of the Δ*relA* Δ*spoT* double mutant when compared to WT (Supplementary Fig. 8). Thus, the modest effect of the single Δ*relA* deletion and the double Δ*relA* Δ*spoT* deletions on AlpA abundance is unlikely to explain the greater than 1000-times increase in the ability of these cells to tolerate ectopically produced AlpA when compared to WT. (Note that the potential role of *PA4114* in contributing to the toxicity of ectopically produced AlpA was not investigated further.)

To determine whether cells of the PAO1 Δ*relA* mutant strain survive ectopic expression of *alpA* due to a decrease in AlpA activity in these cells, we used RNA-Seq to compare the effects of ectopically produced AlpA in WT and Δ*relA* mutant cells. The results depicted in Supplementary Data 3 and 4 indicate that in general, AlpA appeared to be between 2–3 times more active in WT cells than in cells of the Δ*relA* mutant. Note that for those transcripts that are positively regulated by AlpA and appear to result from bypassing intrinsic transcription terminators, such as

anti-sense *PA1416* and *cyaA* (Fig. 5b, c), the fact that the Δ*relA* mutation does not appear to influence the expression of the gene (s) upstream of the terminator (Supplementary Table 2) argues that RelA influences AlpA activity rather than the activity of the promoter(s) that drive expression of the corresponding operon.

To test further whether RelA enhances the activity of AlpA, we measured the abundance of two AlpA-regulated transcripts by qRT-PCR in cells of a Δ*alpB* Δ*relA* double mutant and in cells of a Δ*alpB* mutant following ectopic synthesis of the AlpR-CTD. (The use of Δ*alpB* mutant cells for this experiment ensured that cells expressing the *alp* genes at high levels would not lyse as readily and would thus still be present in the cell population from which the RNA was isolated.) Figure 6a shows that induction of endogenous AlpA through ectopic synthesis of the AlpR-CTD resulted in an 81-fold increase in expression of *PA0807* in the presence of RelA and a 13-fold increase in expression in the absence of RelA. Similarly, ectopic synthesis of the AlpR-CTD resulted in a 133-fold increase in expression of *alpD* in the presence of RelA and a 69-fold increase in expression in the absence of RelA. Of note, the basal expression of the *PA0807* and *alpD* genes was not positively regulated by RelA (Fig. 6a). These findings support the observations from our RNA-Seq studies that RelA enhances the activity of AlpA.

We reasoned that if RelA and SpoT enhance the activity of AlpA then Δ*relA* Δ*spoT* mutant cells should be less susceptible to AlpA-mediated PCD than their WT counterparts. To test this prediction, we compared the effects of ectopically produced AlpR-CTD on the viability of WT and Δ*relA* Δ*spoT* mutant cells. The results depicted in Fig. 6b indicate that cells of the Δ*relA* Δ*spoT* mutant are less susceptible to AlpA-mediated PCD, when compared to WT cells, by at least two orders of magnitude. These findings suggest that cells of the Δ*relA* Δ*spoT* mutant strain can survive ectopic synthesis of the AlpR-CTD due to a decrease in lysis gene expression, likely as a result of decreased AlpA activity.

To further test the hypothesis that the presence of the small molecule ppGpp causes an increase in the activity of AlpA, we compared expression of an AlpA-regulated reporter gene in cells of *E. coli* that either could or could not produce ppGpp. The AlpA-regulated reporter gene used for these experiments consisted of the *alpB* promoter, followed by two copies of the tR′ terminator, followed by *lacZ*. A reporter containing the *lacUV5* promoter, followed by two copies of the tR′ terminator, followed by *lacZ*, served as a control. The results depicted in Fig. 6c indicate that in the presence of AlpA, the expression of the reporter under the control of the *alpB* promoter (i.e. AlpA-mediated antitermination) was specifically reduced sevenfold in cells of the *E. coli relA spoT* mutant reporter strain when compared to WT cells (that could produce ppGpp). In addition, the results depicted in Supplementary Fig. 9 indicate that expression of an AlpA-regulated reporter gene that was under the control of the *alpB* promoter was reduced twofold in PAO1 Δ*relA* Δ*spoT* mutant cells when compared to WT cells, although the apparent small effect of the Δ*relA* Δ*spoT* mutations on AlpA abundance (Supplementary Fig. 8) presumably accounts for some of the observed effect. Taken together, these findings suggest that in both cells of *P. aeruginosa* and in cells of *E. coli*, AlpA is less active if ppGpp cannot be produced.

One of the key regulatory targets of ppGpp is RNAP. Indeed, in *E. coli*, ppGpp is known to interact with two distinct surfaces on the β′ subunit of RNAP, with the binding of ppGpp to site 1 requiring the ω subunit, and the binding of ppGpp to site 2 requiring DksA[30–32]. In order to test whether the binding of ppGpp to these sites on RNAP might modulate the ability of AlpA to function as an antiterminator, we measured AlpA activity in cells lacking either ω or DksA. For these studies, we constructed reporter strains of PAO1 that either contained a stop

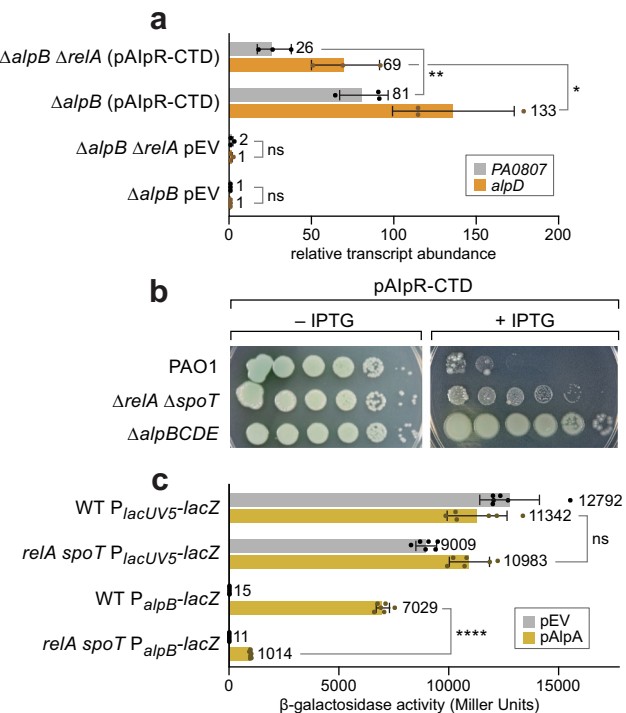

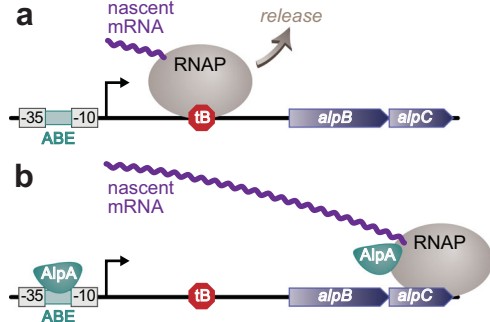

**Fig. 7 Proposed model of how AlpA functions at the *alpB* promoter.**
**a** In the absence of AlpA, transcription termination occurs at intrinsic terminator tB positioned downstream of the *alpB* promoter. Nascent transcript is in purple and RNAP is in gray. **b** AlpA first binds promoter at the putative AlpA binding element (ABE) positioned between the −10 and −35 elements of the promoter, then binds RNAP to allow RNAP to bypass the tB intrinsic terminator positioned downstream.

**Fig. 6 ppGpp modulates AlpA activity in *P. aeruginosa* and in *E. coli*.**
**a** qRT-PCR was used to measure transcript abundance of genes in the two AlpA-regulated operons in strains that have *alpB* deleted or *alpB* and *relA* deleted. ΔalpB was used as the background strain to prevent lysis of cells ectopically synthesizing the AlpR-CTD. In Δ*relA* mutant cells that contain the *alpR-CTD* expression vector pAlpR-CTD, there is a threefold decrease in transcript abundance of *PA0807* (gray) and twofold decrease in transcript abundance of *alpD* (gold) compared to WT (i.e. *relA*+) cells. Values and error bars reflect mean ± SD of n = 3 biological replicates. Two-tailed, unpaired, unequal variance *t*-tests were used to calculate *p*-values between indicated samples. ΔalpB ΔrelA (pAlpR-CTD) vs. ΔalpB (pAlpR-CTD) for the *PA0807* transcript: *p* = 0.0078. ΔalpB ΔrelA (pAlpR-CTD) vs. ΔalpB (pAlpR-CTD) for the *alpD* transcript: *p* = 0.019. *p*-Values indicated by the following symbols: >0.05 = ns, ≤0.05 = *, ≤0.01 = **. **b** Image shows tenfold serial spot dilutions of cells of indicated strains of *P. aeruginosa* (PAO1 is WT) ectopically expressing the *alpR-CTD* under the control of an IPTG-inducible promoter from plasmid pAlpR-CTD. Ectopic expression of *alpR-CTD* is lethal due to activation of *alp* encoded PCD pathway. Deletion of *alpBCDE* suppresses lethality of ectopic *alpR-CTD* expression. PAO1 cells with *relA* and *spoT* deleted survive ectopic expression of *alpR-CTD* better that WT cells but not as well as cells of the Δ*alpBCDE* mutant strain. **c** β-galactosidase activity (in Miller Units) of the indicated reporter constructs in *E. coli*. Promoters were either *lacUV5* or *alpB* and strain backgrounds were either WT and could produce ppGpp (indicated WT) or were *relA spoT* mutants that cannot make ppGpp (indicated *relA spoT*). Cells contained plasmid pAlpA (gold) or plasmid pEV (the empty vector; gray). Values and error bars reflect mean ± SD of n = 3 biological replicates in technical duplicate. Two-tailed, unpaired, unequal variance *t*-tests were used to calculate *p*-values between indicated samples. WT P*alpB* (pAlpA) vs. *relA spoT* P*alpB* (pAlpA): *p* = 7.6 × 10−8. *p*-Values indicated by the following symbols: >0.05 = ns, ≤0.0001 = ****. Source data are provided as a Source Data file.

codon early on in the *rpoZ* gene (encoding ω), or that contained deletions of the two genes encoding DksA orthologs in *P. aeruginosa*. The results depicted in Supplementary Fig. 10 indicate that the loss of the ω subunit of RNAP did not appreciably influence the ability of AlpA to promote expression of an *alpB* promoter-*lacZ* fusion, nor did the loss of the two DksA

orthologs appreciably influence the ability of AlpA to promote expression of either an *alpB* promoter-*lacZ* fusion or a *PA0807* promoter-*lacZ* fusion. These findings suggest that ppGpp is unlikely to potentiate the activity of AlpA through its interaction with site 1 or site 2 of RNAP.

## Discussion

Here we demonstrate that in *P. aeruginosa*, AlpA positively regulates a PCD pathway, as well as genes in a second newly identified locus (*PA0807–PA0829*), by functioning as a processive antiterminator. We show further that specific sequences at target promoters are important for AlpA activity, that AlpA can interact with two distinct subunits in RNAP, and present evidence that the activity of AlpA may be potentiated by the alarmone ppGpp. Our findings support a working model in which AlpA recognizes specific sequences within target promoters, enabling the regulator to load onto RNAP at these locations; when bound by AlpA, RNAP acquires the ability to bypass transcription terminators located upstream of the first gene in each of two target operons (Fig. 7). Through its apparent responsiveness to ppGpp, AlpA may enable the integration of environmental cues into the decision to execute a PCD pathway.

AlpA is thought to contribute to the virulence of *P. aeruginosa* in an acute lung infection model by switching on expression of the *alpBCDE* self-lysis genes in a subset of cells[12]; the lysis of a subset of cells in the population appears to result in the release of a factor(s) that enhances the survival of the remainder of the population[12]. The AlpA-regulated genes in the *PA0807–PA0829* locus do not appear to contribute to cell lysis but several of them could contribute to survival in the host. Indeed, *PA0807* (*ampDh3*) encodes a peptidoglycan remodeling factor that has been shown to be important for virulence[13]. Moreover, the uncharacterized *PA0829* gene has previously been implicated as important for virulence in a rat chronic lung infection model[14]. Release of proteins encoded in this second AlpA-regulated locus by lysis of a subset of *P. aeruginosa* cells could enhance survival of the rest of the population in the host. We speculate that by controlling both the *alpBCDE* cell lysis genes and the *PA0807–PA0829* locus, AlpA might provide an efficient means for coordinating both the production and release of factors into the host that facilitate the survival of other *P. aeruginosa* cells.

We have identified a DNA sequence that is important for regulation of target genes by AlpA. This putative ABE is located between the −10 and −35 elements of the *alpB* and *PA0807* promoters. Notably, the location of the putative ABE is similar to

the location of the binding element for the well-characterized processssive antiterminator λQ at the bacteriophage λ PR′ promoter[23,33]. Like the Q binding element at PR′, which helps direct the loading of Q onto RNAP at this location[34,35], the putative ABEs at the *alpB* and *PA0807* promoters could help to direct the loading of AlpA onto RNAP bound at these specific genomic locations.

We have found that AlpA can interact with a portion of RNAP referred to as the β-flap. Although the importance of the interaction between AlpA and this region of RNAP for antitermination has not been determined, the antiterminators λQ, bacteriophage λ N, P7 (from a *Xanthomonas oryzae* phage) and Gp39 (from a *Thermus thermophilus* phage) are also known to interact with the β-flap[22,36,37]. While AlpA and λQ do not appear to exhibit any sequence homology, it is possible they function through a similar mechanism. The β-flap is located near the RNA exit channel and recent structural studies of Q from bacteriophage 21 (Q21) interacting with *E. coli* RNAP showed that Q21 principally acts by inserting into the RNA exit channel of RNAP, restricting the size of the exit channel and extending its length[34,35]. Q21 thus acts as a molecular nozzle to prevent formation of terminator hairpin structures that would otherwise form in the nascent RNA[34,35]. We note that sufficiently high ectopic synthesis of AlpA results in RNAP bypassing predicted intrinsic terminator sequences at many locations in the genome, suggesting that AlpA may be able to load onto RNAP in solution and bypass the requirement for an ABE. If AlpA can load onto RNAP that has already synthesized a relatively long nascent transcript, then AlpA could not act in precisely the same fashion as Q21, which could not plausibly thread onto a nascent RNA that has already emerged from the exit channel. Nevertheless, it is possible that AlpA might load early even if it bypasses the requirement for an ABE. We also note that the apparent widespread readthrough of transcription terminators we observe upon high-level ectopic synthesis of AlpA may explain why AlpA is toxic to the cell under these conditions even when the *alpBCDE* cell-lysis genes are absent (Fig. 5a). Indeed, widespread readthrough of transcription terminators is thought to account for the toxic effects of depleting Rho in *Mycobacterium tuberculosis*[38].

We present evidence that AlpA can interact with a second region of RNAP; region 1.1 of σ70. This region of σ70 is thought to prevent free σ70 from interacting with promoter DNA, but it is also involved in open complex formation and in the RNAP holoenzyme structure is positioned within the active site channel[26,27,39]. Q is known to interact with σ70 but it interacts with region 4, not region 1.1; the interaction between λQ and region 4 of σ70 stabilizes the formation of a paused complex on the DNA that facilitates the loading of λQ onto RNAP[40]. It is unclear how an interaction between AlpA and region 1.1 of σ70 could influence termination and we cannot rule out the possibility that the interaction we detect between AlpA and region 1.1 of σ70 in our two-hybrid assay is due simply to the fact that this region of σ70 is negatively charged. We note that the only other regulator known to interact with region 1.1 of σ70 is Gp2 of bacteriophage T7, which acts to inhibit transcription initiation in *E. coli* by preventing this portion of σ70 from exiting the active site channel, thus preventing entry of the promoter DNA[41]. We find that sufficiently high ectopic synthesis of AlpA in *P. aeruginosa* not only promotes the synthesis of certain sense and anti-sense transcripts by causing widespread terminator readthrough, but also represses the expression of hundreds of genes. Further work will be required to determine whether an interaction with region 1.1 of σ70 plays any role in AlpA-mediated antitermination and whether any of the observed repressive effects of AlpA might be explained through this potential interaction.

Lastly, we have obtained evidence suggesting that the activity of AlpA is stimulated, either directly or indirectly, by the small molecule ppGpp. ppGpp is present in almost all bacterial species and is produced in response to various starvation conditions in *E. coli* and *P. aeruginosa* by RelA and SpoT. Recently it has been shown that in *E. coli*, ppGpp is produced in response to DNA damage[42]. If AlpA is expressed but ppGpp is not present, we see a decrease in expression of AlpA-regulated genes (i.e. a decrease in antitermination) and consequently less PCD (Fig. 6a, b). Modulation of AlpA activity by ppGpp could provide a failsafe mechanism to ensure the PCD pathway is activated only in response to DNA damage, or could provide a mechanism for modulating the proportion of cells in which PCD is activated through environmental cues that alter ppGpp abundance. Although AlpA appears to be the only antiterminator whose activity might be potentiated by ppGpp[43], several phage-encoded antiterminators are actively produced in response to DNA damage as a result of prophage induction, and so it will be interesting to determine whether or not ppGpp enhances the activity of AlpA by binding AlpA directly (and possibly modulating AlpA's interactions with the DNA or with RNAP), and also whether other antiterminators are responsive to this alarmone.

AlpA is the only virulence gene regulator we are aware of in *P. aeruginosa* that acts as a processive antiterminator. Although in enteric bacteria orthologs of the antiterminator RfaH (a NusG-related protein) control the expression of virulence genes, *P. aeruginosa* does not appear to encode a RfaH ortholog[44]. The Alp system is encoded by the bacterial chromosome but bears features characteristic of prophage genes. For example, *alpB* and *alpC* encode a putative holin and anti-holin, respectively, whose orthologs contribute to host cell lysis and are typically found in certain prophages[12]. The AlpA regulator itself, as we show here, also bears functional similarities to phage-encoded regulators such as the Q protein from bacteriophage λ[17,22,45]. The Alp system, therefore, likely originated from a prophage and was repurposed to provide a bacterial self-lysis system that may be responsive to both DNA damage and ppGpp.

## Methods

**Bacterial strains and growth conditions**. All strains used in this study are listed in Supplementary Table 3. *E. coli* strains CSH100 and FW102 were used to create *E. coli* reporter strains containing promoters driving the expression of *lacZ*[46]. *E. coli* strain SM10 (λpir) was used for conjugal transfer of plasmids into *P. aeruginosa*, *E. coli* strain FW102 OL2-62 was used for the bacterial two-hybrid assays[22]. For introducing in-frame deletions by conjugal transfer, *P. aeruginosa* strain PAO1 was selected on Pseudomonas isolation agar (PIA) and grown on low-salt LB agar supplemented with 5% (wt/vol) sucrose at 37 °C for *sacB* counter-selection. *E. coli* strains were supplemented with 15 μg/mL gentamicin, 10 μg/mL tetracycline, 100 μg/mL carbenicillin, 30 μg/mL kanamycin, and 25 μg/mL chloramphenicol, as needed. *E. coli* strains that contained plasmids with IPTG-inducible promoters were supplemented with the indicated IPTG concentration. For *P. aeruginosa*, gentamicin (LB: 30 μg/mL; PIA: 60 μg/mL) and tetracycline (LB: 35 μg/mL; PIA: 200 μg/mL) were added as needed. Liquid cultures of *P. aeruginosa* were inoculated at a starting OD600 of 0.01 and grown with aeration at 37 °C in LB broth. For induction of IPTG-inducible promoters in liquid-grown *P. aeruginosa* cultures, IPTG was added at a final concentration of 1 mM. For induction of IPTG-inducible promoters in liquid-grown *E. coli* cultures, IPTG was added at a final concentration of 5 μM.

**Induction of DNA damage**. Cultures were grown to an OD600 of 0.5 and ciprofloxacin (Sigma-Aldrich) was added at a final concentration of 1 μg/mL for the specified times before harvesting. For microscopy, cells were grown to an OD600 of ~0.3–0.5 in LB with 30 μg/mL gentamicin, then IPTG was added at a final concentration of 5 mM and cells were grown for an additional 30 min. Cells were then mixed and spotted on agarose pads containing 5 mM IPTG and imaged in the phase channel as described in the "Time-lapse microscopy" section.

**DNA manipulations**. Standard molecular cloning procedures were followed. Oligonucleotide primers were obtained from Sigma Life Sciences. DNA amplification was performed using KOD polymerase (Novagen). DNA sequencing was performed by Genewiz Inc. Restriction enzymes were obtained from New England Biolabs.

**Plasmid construction**. All plasmids and primers used in this study are listed in Supplementary Tables 3 and 4. The suicide vector pEXG2 (ref. [47]) was used to make pEXG2-AlpA-TAP, pEXG2-P*alpB*ABE6, P*PA0807*ABE7, and pEXG2-*rpoZ* (Stop). pEXG2 (ref. [47]) was also used to make the in-frame deletion constructs pEXG2-ΔPA0807-PA0831, pEXG2-Δ*dksA*, pEXG2-Δ*dksA2*, pEXG2-Δ*relA*, and pEXG2-Δ*spoT*. The pEXG2 vector was also used to introduce point mutations into the putative ABEs of the endogenous *alpB* and *PA0807* promoter regions as well as introducing an early stop codon into *rpoZ*. For generation of the P*alpB* ABE6 and P*PA0807* ABE7 mutant promoters and *rpoZ*(Stop), overlap extension PCRs were performed with oligonucleotides that introduced the six point mutations into the *alpB* promoter region, seven point mutations in *PA0807* promoter region and an early stop codon in the *rpoZ* ORF, respectively. The resulting PCR products were combined via overlap extension PCR, digested with the appropriate restriction enzymes and cloned into pEXG2. The pEXG2-AlpA-TAP insertion vector was created by PCR amplifying AlpA and TAP sequences using primers with restrictions sites. A three-way ligation was then performed using AlpA PCR product, TAP PCR product, and pEXG2 backbone. In-frame deletion plasmids were constructed by amplification of ~400 bp regions of genomic DNA flanking the gene to be deleted, with primers containing restriction sites, followed by digestion and three-way ligation into pEXG2 that had been digested with the appropriate restriction enzymes. In addition to harboring flanking DNA, the pEXG2-Δ*relA* and pEXG2-Δ*spoT* deletion plasmids contained the original start and stop codon of *relA* and *spoT*, respectively, separated by DNA specifying three alanine codons.

All P*alpB*-*lacZ* F′ episome *lacZ* reporter plasmids were produced by amplification of PAO1 genomic DNA and ligation of the *alpB*, and *PA0807* promoter regions into pFW11 (ref. [46]). Point mutations in the P*alpB* promoter regions of the *lacZ* reporters were introduced using overlap extension PCR.

The plasmid pMini-CTX-lacZ[48] was used to make *lacZ* reporter fusions to the promoter regions of *alpB* and *PA0807*, by amplification of promoter regions by PCR, restriction digested with EcoR1 and BamH1 and ligation into digested pMini-CTX-lacZ.

For the bacterial two-hybrid assays, PAO1 DNA specifying RNAP β-flap (836–1074), σ[70] (1–58), σ[70] (1–95), and AlpA were amplified from genomic DNA and the PCR products were then ligated in-frame into the vectors pACλCI32 and pBRαLN[49], resulting in pACλCI-Paβflap, pACλCI-Paσ[70] (1–58), pACACI-Paσ[70] (1–95), and pBRα-AlpA, respectively. Plasmids were introduced into *E. coli* strain FW102 F′::placOL2-62-LacZ[22].

The plasmid pPSV38 (ref. [50]) harboring AlpA (pAlpA) has been described previously[12] and was used to create plasmid pAlpA-V that synthesizes AlpA with VSV-G (vesicular stomatitis virus glycoprotein-G) epitope tag fused to its C-terminus (AlpA-V).

**Strain construction**. For pEXG2-based vectors, standard allelic replacement was performed to introduce the desired mutations[47,51]. Briefly, *E. coli* SM10 cells carrying the plasmid were mated with recipient *P. aeruginosa* cells. Primary integrants were selected on PIA containing 60 µg/mL gentamicin. The integrated plasmids were resolved by growth on low-salt LB agar plates containing 5% sucrose for *sacB* counterselection. Sucrose-resistant colonies were then screened for loss of gentamicin resistance. Sucrose-resistant and gentamicin-sensitive clones were then screened for the desired mutation by PCR. For introduction of point mutations, colonies were screened by sequencing of PCR products of the corresponding mutated genomic regions.

The *E. coli* strain FW102 Δ*relA* *spoT*::*cat* was constructed using the lambda red cloning system[52]. Briefly, the FRT-flanked chloramphenicol resistance gene was PCR amplified using the primers pKD3relA-del and pKD3-spoTdel_F2 and R2 primers, which contain *spoT* homology sequences. This PCR product was digested with Dpn1 then electroporated into FW102 Δ*relA* strains containing plasmid pKD46. Transformants were selected on LB containing 15 µg/mL chloramphenicol and re-streaked on LB-agar plates containing carbenicillin (15 µg/mL), to ensure loss of pKD46. Mutations were confirmed with colony PCR using col_catF2 and col_spoT_R2 primers.

The miniCTX P*alpB* and *PA0807 lacZ* fusion reporter constructs were mobilized from *E. coli* SM10 into the appropriate recipient *P. aeruginosa* strains (PAO1, Δ*relA* Δ*spoT*, *rpoZ*(stop), and Δ*dksA* Δ*dksA2*). Transconjugants were selected on PIA agar containing 200 µg/mL tetracycline.

The F′ P*alpB* and P*lacUV5 lacZ* fusion constructs were mobilized from *E. coli* CSH100 into *E. coli* FW102 and FW102 Δ*relA* *spoT*::*cat*. Transconjugants were selected on LB containing streptomycin 100 µg/mL and kanamycin 50 µg/mL.

**RNA-Seq and data analysis**. For RNA-Seq comparing PAO1 WT, PAO1 *alpA* (stop), and PAO1 Δ*alpA* mutant cells, overnight cultures of biological triplicate samples were back-diluted to OD$_{600}$ of 0.01 in 3 mL of LB and grown to mid-log-phase (OD$_{600}$ of ~0.5). Ciprofloxacin was then added to cultures at a final concentration of 1 µg/mL and 1 mL of each culture was harvested 120 min later. TriReagent was used for RNA isolation (Molecular Research Center). RNA isolation was performed using Zymo Direct-zol RNA Miniprep Plus kit according to kit instructions. RNA was then sent to the Broad Institute for library prep and sequencing or was made into cDNA for qRT-PCR analysis. RNA-Seq libraries were constructed and sequenced at the Broad Institute of MIT and Harvard by the Microbial 'Omics Core and Genomics Platform, respectively. Sequencing reads

from each sample were demultiplexed based on their associated barcode sequence using custom scripts. Reads were aligned to the PAO1 genome (NC_002516) using BWA (version 0.7.15)[53] and read counts were assigned to genes and other genomic features using custom scripts. Differential expression analysis was conducted with DESeq2 (ref. [54]).

For RNA-Seq experiments comparing PAO1 containing plasmid pAlpA to PAO1 containing plasmid pPSV38 (the empty vector control; referred to as pEV) and PAO1 Δ*relA* containing plasmid pAlpA to PAO1 Δ*relA* containing plasmid pPSV38, biological triplicate samples of cells were back-diluted from overnight cultures to an OD$_{600}$ of 0.01 in 200 mL of LB supplemented with 30 µg/mL gentamicin and grown at 37 °C with shaking for 2 h (to an OD$_{600}$ of ~0.04). IPTG was then added at a final concentration of 1 mM and cells were grown for another 90 min to mid-log-phase (i.e. an OD$_{600}$ of ~0.3–0.4). Specifically, the OD$_{600}$ of the cultures used for RNA isolation were as follows: WT with pAlpA (0.36–0.38), WT with pEV (0.36–0.42), Δ*relA* with pAlpA (0.3–0.34), and Δ*relA* with pEV (0.3–0.4); 2 mL of cells were harvested by centrifugation at 3200×*g* for 10 min. RNA was isolated by resuspending cells in 1 mL of TriReagent (Molecular Research Center) and cells were lysed by incubation at 60 °C for 10 min[50]. Supernatants were extracted with 200 µL of chloroform, RNA was then precipitated with ethanol, pelleted by centrifugation at 21,000×*g* and washed with 75% ethanol[50]. Following resuspension in water, RNA was treated with RNase-free DNase I (Promega) for 1 h at 37 °C, then purified through a second round of treatment with TriReagent and chloroform[50]. Following a final ethanol precipitation step, RNA was resuspended in water. RNA-Seq libraries were made by first depleting ribosomal RNA from 5 µg of total RNA using the Ribo-Zero Magnetic Kit (Bacteria) (Epicentre) according to the manufacturer's specifications. The remaining RNA was ethanol-precipitated, then used to generate RNA-Seq libraries using the NEB-Next Multiplex Small RNA Library Prep Kit for Illumina (New England Biolabs) according to the manufacturer's protocol. Libraries were size-selected by PAGE using a 5% gel with TBE (Biorad), stained with SYBR gold nucleic acid stain (Life Technologies), and visualized using a blue-light transillumination system. Fragments corresponding to 150–300 bp were gel purified, ethanol-precipitated, and resuspended in TE buffer according to the NEBNext Multi-plex Small RNA Library Prep Kit for Illumina (New England Biolabs) protocol[50]. RNA quality was determined by Agilent Bioanalyzer. Libraries were sequenced by Elim Bio-pharmaceuticals, Inc. (Hayward, CA), using an Illumina HiSeq 2500 which generated 50 bp single-end reads. Reads were aligned to PAO1 genome (NC_002516) using Bowtie2 (ref. [55]). Differential expression analysis was conducted with DESeq2 (ref. [54]).

**RNA isolation and qRT-PCR**. PAO1 and PAO1 *alpA*(Stop) cells were back-diluted from overnight cultures to OD$_{600}$ of 0.01 in 3 mL of LB and grown to mid-log-phase (OD$_{600}$ of ~0.5). Then ciprofloxacin was added to cultures at a final concentration of 1 µg/mL and samples were harvested 120 min later. *E. coli* F′ P*alpB*-*lacZ* cells were back-diluted from overnight cultures to an OD$_{600}$ of 0.01 in 3 mL of LB supplemented with 15 µg/mL gentamicin, 50 µg/mL kanamycin, and 5 µM IPTG and grown to mid-log-phase (OD$_{600}$ of ~0.5); 1 mL of each culture was harvested. TriReagent was used for RNA isolation (Molecular Research Center). RNA isolation was performed using Zymo Direct-zol RNA Miniprep Plus kit according to kit instructions. cDNA synthesis using SuperScript III reverse transcriptase (Invitrogen) and qRT-PCR were performed using FastStart Essential DNA Green Master (Roche) and a LightCycler 96 system (Roche). The abundances of transcripts were measured relative to the abundance of the *clpX* transcript. qRT-PCR was performed at least twice on sets of biological triplicate samples. Relative expression values were calculated by using the comparative threshold cycle ($C_T$) method ($2^{-\Delta\Delta CT}$)[56]. The fold enrichment values shown are the means from three biological replicates, and error bars represent the standard deviation of the mean. The data shown are from one representative experiment.

**ChIP-Seq library preparation and ChIP-qPCR**. The ChIP-Seq experiment was done as described[12]. Briefly, we performed ChIP-Seq in PAO1 pPSV37-AlpA-TAP and PAO1 pPSV37-AlpA (used for the mock immunoprecipitation control) using biological triplicate samples. We grew diluted cells from overnight cultures to mid-log in 200 mL of LB supplemented with gentamicin (30 µg/mL final concentration) with shaking at 37 °C. Upon collection, we immediately cross-linked cells with formaldehyde (1% final concentration) for 30 min, followed by quenching of the cross-linking reaction with 250 mM glycine. Next, we lysed the cells and sheared the DNA with a Bioruptor water bath sonicator (Diagnode). We combined the lysates with anti-IgG beads (GE Healthcare) for IP. After washing the samples, we reversed the cross-linking by incubating the samples at 65 °C overnight. We isolated DNA with a PCR purification kit (Qiagen) and determined DNA yields using a Nanodrop or Agilent Bioanalyzer. Samples were sequenced by Elim Biopharmaceuticals, Inc (Hayward, CA). We performed ChIP-qPCR on triplicate cultures grown from independent colonies and cells were lysed using a tip sonicator.

**ChIP-Seq data analysis**. Read mapping and peak calling was essentially as described previously[57]. In brief, reads were mapped to the PAO1 genome (NC_002516.2) and the expression plasmids pPSV37-AlpA-TAP and pPSV37-AlpA using bowtie2-2.0.6 (ref. [55]) allowing up to one mismatch per seed. All the mock IP (PAO1 with the vector pPSV37-AlpA) replicate data were merged and

used as "background" for each biological replicate (PAO1 with the vector pPSV37-AlpA-TAP). The program QuEST (version 2.42)[58] was used to call peaks. Regions in each biological replicate were considered peaks if they are 2.5-fold enriched for reads over background, have a positive peak shift and strand correlation, and have a $q$-value of <0.01. AlpA enrichment peaks are defined as the minimal region identified in at least two biological replicates. Tracks were visualized using the Integrative Genomics Viewer (IGV) version 2.3 (ref. [59]).

**Quantitative PCR**. qPCR was performed on DNA isolated from ChIP experiments using FastStart Essential DNA Green Master (Roche) and a LightCycler 96 (Roche). Primer efficiencies were calculated by melting curve analyses. Data analyses were supported by LightCycler software version 1.1 (Roche). For ChIP, relative fold enrichment indicates the relative abundance of a DNA region of interest relative to a negative control region (in the gene $clpX$) and the amount of DNA in the input. Specifically, we calculate fold enrichment = $1.9^{\Delta\Delta Ct}$, $\Delta\Delta Ct$ = (Ct_ChIP$_{clpX}$ – Ct_ChIP$_{target}$) – Ct_Input$_{clpX}$ – Ct_Input$_{target}$). Reported fold enrichments are the mean of three biological replicates, and error bars denote SD. All data shown are representative of at least two independent experiments.

**β-Galactosidase assays**. LacZ transcriptional reporter assays and bacterial two-hybrid assays were performed by first permeabilizing cells with CHCl$_3$ and SDS[25]. o-nitrophenyl-β-D-galactoside (ONPG) was then added to each sample to assay for β-galactosidase activity[25]. For the two-hybrid assays reported in Fig. 4b, assays were performed with cells grown in LB supplemented with IPTG (5 μM). Values and error bars represent the mean ± SD of $n = 3$ biological replicates in technical duplicate. β-Galactosidase experiments were performed at least twice. For bacterial two-hybrid screens with the E. coli RNAP sigmaome and coreome (Supplementary Fig. 7), 96-well plate β-galactosidase assays were performed in media supplemented with IPTG (20 μM)[60,61].

**AlpR sequestration assays on LB agar plates**. For AlpR sequestration assays[12], the pAlpR-CTD plasmid, synthesizing the AlpR-CTD, was introduced into the indicated strain of PAO1 by electroporation. Colonies of plasmid-containing cells were selected on LB agar containing gentamicin (30 μg/mL) and resuspended in PBS to an OD$_{600}$ of 0.01. Tenfold serial dilutions of cells (10 μL) were spotted onto LB agar plates containing gentamicin with or without 10 mM IPTG. Plates were incubated overnight at 37 °C before being photographed.

**Time-lapse microscopy**. Time-lapse microscopy sequences were acquired on a Nikon Ti inverted microscope with a 100× oil objective, automated focusing (Perfect Focus System, Nikon), a Lumencore SpectraX LED illumination, a sCMOS camera (Andor Zyla 4.2 Plus), and Nikon Elements 4.30 acquisition software (Nikon). PAO1 mCherry (pAlpR-CTD) and PAO1 GFP P$_{alpB}$ABE6 (pAlpR-CTD) were back-diluted from overnight cultures to OD$_{600}$ of 0.01 and grown to mid-log phase. IPTG was added at a final concentration of 5 mM. Cells were concentrated 5×, mixed and spotted on LB agarose supplemented with gentamicin 30 μg/mL and 5 mM IPTG. Images were acquired every 5 min for 6 h at 37 °C.

**Western blot analyses**. Whole-cell lysates from biological triplicates were separated by SDS-PAGE on 4–12% Bis-Tris NuPAGE gels in MES running buffer (Thermo Fisher). Proteins were transferred to polyvinylidene fluoride (PVDF) membranes with the XCell-II Blot Module (Thermo Fisher). Membranes were blocked overnight with Intercept (PBS) Blocking Buffer (LI-COR). Membranes were probed with antibodies that recognize the VSV-G epitope tag on AlpA-V (Anti VSV-Glycoprotein antibody, mouse monoclonal, SigmaAldrich) used at a dilution of 1:3333, or with antibodies that recognize the α-subunit of RNAP (anti-E. coli RNAP α antibody, BioLegend) used at a dilution of 1:5000. Membranes were re-blocked and incubated with near-Infrared secondary antibody IRDye 680LT donkey anti-mouse IgG (LI-COR) used at a dilution of 1:30,000. Imaging was performed on an Azure c600 Imager, and fluorescence intensity was quantified using Image Studio software (LI-COR).

**Transposon mutagenesis and identification of genomic regions containing transposon insertions**. For transposon mutagenesis, WT cells of P. aeruginosa strain PAO1 were first mutagenized with a version of the mariner transposon that confers resistance to tetracycline[62], then transformed with plasmid pAlpA. Transformants were then selected following growth on LB agar plates containing IPTG (5 mM). The identification of the location of transposon insertion sites in mutants that tolerated growth on LB agar plates containing IPTG was determined by arbitrary PCR followed by DNA sequencing[62].

**Reporting summary**. Further information on research design is available in the Nature Research Reporting Summary linked to this article.

## Data availability

The RNA-Seq and ChIP-Seq data supporting this study are deposited in the National Center for Biotechnology Information Gene Expression Omnibus under accession number GSE152485 and are available at https://www.ncbi.nlm.nih.gov/geo/query/acc.cgi?acc=GSE152485. All other data supporting the findings of this study are available within the paper, within the Supplementary Data or Supplementary Information, or are available upon request. Source data are provided with this paper.

## Code availability

The custom scripts used for demultiplexing RNA-Seq reads based on their associated barcode sequence are available at https://github.com/broadinstitute/split_merge_pl. Custom scripts used for assigning RNA-Seq read counts to genes and other genomic features are available at https://github.com/broadinstitute/BactRNASeqCount.

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

## Acknowledgements

We thank members of the Dove lab for discussions, Renate Hellmiss for artwork, Ashley Cronin for help with the experiments in Supplementary Fig. 7, Joseph Mougous for help with Supplementary Fig. 5, and Ann Hochschild for discussions and comments on the manuscript. We also thank Paula Montero Llopis at the Microscopy Resources on the North Quad (MicRoN) core facility at Harvard Medical School for help with time-lapse microscopy. Certain RNA-Seq libraries were constructed and sequenced at the Broad Institute of MIT and Harvard by the Microbial 'Omics Core and Genomics Platform, respectively. The Microbial 'Omics Core also provided guidance on experimental design and conducted preliminary analyses for certain RNA-Seq data. We thank Jonathan Livny for help with providing access to the scripts for RNA-Seq data analyses. This work was supported by NIH grant AI118955 (to S.L.D), Howard Hughes Medical Institute Gilliam Fellowship and NSF graduate student research fellowship (to J.M.P.), and NSF grant RUI 1714103 (to P.D.).

## Author contributions

J.M.P., S.M.P., K.A.M., T.K.K., and P.D. performed the experiments. K.R.M. carried out the data analysis. J.M.P. and S.L.D. wrote the manuscript. S.L.D. supervised the study. Funding to support the work was acquired by P.D. and S.L.D.

## Competing interests

The authors declare no competing interests.
