## [Peer Review File · Nature Communications]

REVIEWER COMMENTS

Reviewer #1 (Remarks to the Author):

In this study, the authors analyze transcription regulation by the regulatory protein AlpA from *P. aeruginosa* (PAO1) and conclude that it acts as a processive transcription antiterminator that binds RNAP at the promoter and then travels with the elongation complex and allows read-through of intrinsic transcription terminators. It is shown that deletion of AlpA results in inactivation of transcription from two specific groups of genes under stress conditions, and that AlpA allows terminator read-through in reporter constructs *in vivo*. Putative promoter and AlpA-responsive elements are further identified in these genes; substitutions of these elements are shown to strongly decrease AlpA effects on reporter gene expression. A two-hybrid screen is then used to identify possible RNAP domains involved in interactions with AlpA. The authors further show that overexpression of AlpA results in dramatic changes in the transcriptome of PAO1, with many read-through transcripts appearing. This suggests that AlpA can act as a non-operon-specific transcription antiterminator *in vivo*, if expressed at high levels. Finally, it is demonstrated that genes involved in biosynthesis of the alarmone ppGpp (*relA* and *spoT*) modulate the effects of AlpA on transcription and cell survival. However, these effects can not be explained by the effects of ppGpp on RNAP properties, since they are not changed upon deletion of *rpoZ* or *dskA*. Overall, this is an interesting system of a transcription antiterminator in PAO1 that may act similarly to phage-encoded processive antiterminators in *E. coli*. However, many conclusions are based on indirect evidence obtained *in vivo* and should be confirmed by biochemical studies *in vitro* with purified proteins, AlpA and RNAP from PAO1. Furthermore, structural analysis of the complexes of AlpA with RNAP and DNA could greatly increase the significance of this work. In the absence of a detailed analysis of the antitermination mechanism, this system remains just another example of transcription regulation, which looks similar to previously analyzed systems – even if it is important for pathogenicity.

Specific comments:

- 1) Lines 94-97 – The putative operon activated by AlpA (from PA0807 to PA0830) contains genes in both orientations and only +strand genes are activated. Is anything known about the actual operon structure of this region? Does it contain any additional promoters, or all transcription starts only upstream of 0807? How activation of transcription from the +strand affects transcription of all the genes encoded in the opposite orientation in the –strand?
- 2) Line 114 and below – The ChIP-seq experiments were apparently performed under different conditions from the RNA-seq experiments (in the absence and in the presence of ciprofloxacin) – what are the reasons and whether this could have any effects on the results obtained?
- 3) Lines 161-163 – The results of the structural modeling should be included in the figures, if they are discussed in the manuscript.
- 4) Lines 163-165 – The fact that AlpA is associated with promoter regions (and downstream of them) by itself cannot prove that AlpA is a DNA binding protein, as well as the presence of a predicted DNA-binding structural motif, as well as the presence of a specific DNA motif required for the AlpA action. Although it is plausible that AlpA may indeed interact with DNA in complex with RNAP, direct DNA binding experiments with purified proteins (AlpA and RNAP) could be more conclusive in this case.
- 5) The effects of substitutions at position -14 could have been explained in more detail in the text. As is evident from Supplementary Fig. 5c, nucleotide substitution at position -14 (PalpB -14*) does not have any effect on promoter activity, while the same substitution in PA0807 has some additive effects to the substitutions from -20 to -25 (Suppl Fig. 5d). Is this nucleotide proposed to interact with AlpA or RNAP?
- 6) Lines 221-240 – Two-hybrid experiments suggest that AlpA may interact with the flap domain and region 1.1 of the sigma70 subunit. However, there are several caveats that make their interpretation difficult:

- The experiments were performed in only one 'orientation' of the AlpA and RNAP portions in the two-hybrid constructs; reciprocal variants should definitely be tested (in which AlpA is fused to CI and RNAP fragments to alpha CTD).
- The results of a two-hybrid screen cannot be interpreted as a final evidence in support of such interactions, since this method can easily give artefactual signals. While interactions with the flap domain seem plausible (but still should be better confirmed by alternative methods), the interactions with the region sigma 1.1 are doubtful. This region is highly negatively charged and largely unstructured and therefore any DNA binding protein could potentially interact with it. It also remains unexplained in the manuscript why the shorter fragment (1-56) binds better than the longer one (1-93) (see also below)
- The same experiments should be done with *Pseudomonas* constructs (at least for verification of the observed interactions with the flap domain and region 1.1)
- Beta' 370-417 seems to interact with AlpA (Suppl Fig. 6A, 4-5-fold enhancement) – why it is not included in further analysis?
- For some reasons, the numbers of beta-gal activity (Miller units) are different for the same measurements shown in Fig. 4B and Suppl. Fig. 6.

7) The only evidence in support of AlpA interactions with sigma region 1.1 is the two-hybrid screen which is not very conclusive (see above). In the absence of any additional experiments confirming these interactions, any speculations about their possible functional role seem groundless. This interaction may also be difficult to explain from the structural point of view (flap and region 1.1 are expected to be positioned from different sides of the DNA binding cleft). Furthermore, region 1.1 is poorly conserved, and the experiments were performed with the *E. coli* sigma, not *Pseudomonas*.

8) Lines 290-292 – The RNA-seq profiles for the *relA* strain should be included in Fig. 5B and C, to illustrate what is said in the text here.

9) Lines 422-423 – It is stated that the Alp system likely originated from a prophage but no evidence in support of this hypothesis is presented in the manuscript. The mechanism of AlpA may indeed be similar to the Q protein of lambdoid phages, but this by itself does not suggest a phage origin (in fact, another antiterminator protein of phage lambda, N appropriated a cellular antitermination complex for its purposes). Does the Alp operon have any features characteristic to prophages, does it contain any known prophage proteins, does it have any differences in the GC content, codon adaptation index etc? What about evolutionary conservation of this system?

10) For all the figures, the explanation of the p-values are missing. There are some numbers of asterisks shown, but they should be clearly explained in each figure. In some cases, the labeling is confusing – for example, in Fig. 2B in the right panel there are four asterisks between 19 and 23 – is this difference significant?

11) Fig. 1A – There are several genes that are significantly activated by the AlpA deletion – what are these genes? Any explanations?

12) Fig. 1A – It seems that the ratio of the mutant to the wild-type cells is shown (since the activity of the AlpA-dependent operons is decreased on the plot), while the figure legend says that it is PAO1 to PAO1 alpA.

13) Fig. 1B – All the genes from the operon together with their orientation should be clearly labeled in the figure.

14) Suppl. Fig. 5C – why the numbers are different from Fig. 3B? (for PalpB Wt and -25 to -20).

Reviewer #2 (Remarks to the Author):

Peña et al. propose that *Pseudomonas aeruginosa* AlpA positively regulates gene expression by

processively antiterminating transcription in a manner analogous to the Q protein of lambdaoid bacteriophages, i.e. it binds to the promoter and travels with RNAP through the operon, preventing termination at intrinsic terminators. The authors identify multiple operons on which AlpA acts, some of which are important for turning on a programmed cell death pathway by activating cell lysis in response to DNA damage. Interestingly, the authors also show that ppGpp enhances antitermination, although the mechanism remains to be worked out. I have a number of questions and minor suggestions the authors may want to consider, but in general, this is a thorough, well-written manuscript that uses ChIP-seq, RNA seq, qPCR, transposase mutagenesis screens, site-directed mutagenesis, and lacZ fusions to identify a heretofore unknown mechanism for regulating gene expression in *Pseudomonas* that integrates environmental responses and virulence. Further biochemical characterization of the mechanism with purified components is probably the next step for determining mechanism. At this point none of the comments below should be considered serious problems.

Comments

1. Line 155: AlpA's ability to function as an anti-terminator does not appear to be dependent upon specific sequence elements within the terminator itself. Isn't this an overstatement, given that the strength of terminators varies, so the ability of AlpA-modified RNAP to cause readthrough might be dependent on how good the terminator is?

2. Line 160-161: In lieu of an actual structure, adding the PHYRE 2 model as a supplementary figure might be helpful

3. Lines 169 and following, Fig 5A, etc: The section on the promoter determinants is not as clear as it might be. For example, explain why the absolute activities of the constructs vary, what the actual substitutions are (change from what residue to what other residue, not just the positions), and which mutations define binding sites of AlpA and AlpB. The definition of the minimal region of the PA0807 promoter region that responds to AlpA might deserve its own paragraph. Probably, there also should be at least one example in which the control and the promoter being tested should be examined using the same terminator.

4. Lines 192-200: The ChIP- qPCR experiment does not really demonstrate that AlpA binds at the position of the mutated sequence (the mutated positions could affect physical contacts nearby). Was any biochemical approach attempted? Does AlpA bind to this region in a footprint or EMSA assay? If no biochemical approaches were attempted to confirm the binding site, then at this point it would be more appropriate to say "putative site".

5. Fig. 4: Was a control performed with the same positive-hit λ CI fusions, but without AlpA fused to RNAP to demonstrate that the effect is dependent on an AlpA interaction with λ CI and not an RNAP-RNAP interaction that bypasses the need for AlpA?

6. Line 346 "that AlpA can interact with two distinct portions of RNAP". Perhaps change "portions of" to "subunits in".

7. Line 400: Is it clear what role an interaction between AlpA and region 1.1 of σ 70 plays in the termination process? Could some of the repressive effects of AlpA observed upon ectopic expression be explained through this interaction? More generally, could AlpA binding reduce transcription by interfering with RPo formation at some promoters when overproduced ectopically?

8. Line 405: "ppGpp is present in all bacteria". This should say "almost all bacterial species" (there are a few examples of bacterial species without detectable ppGpp or a relA gene).

9. Fig. S1 : Since the activities in most cases are very low, the colors of the bars in the histograms are often not visible. Perhaps just a label of some sort would work better.

10. Line 259 : perhaps change "in solution" to "off-DNA"?

11. Lines 321 and 408. This is a semantic issue, but it might help readers understand the mechanism: the lower activity in the absence of ppGpp might be referred to as a reduction in

antitermination (or activation) rather than as a reduction in gene expression.

12. With respect to the mechanism by which ppGpp affects AlpB expression, the ability of ppGpp to affect AlpB-lacZ activity in *E. coli* is an important result. Doesn't this mean that a *P. aeruginosa*-specific factor is not required? If neither Site 1 nor Site 2 on RNAP is required, the authors might want to rule out (e.g. by DRaCALA) or at least mention that ppGpp binds to AlpA itself. Otherwise, perhaps the effect is indirect? Maybe this should be stated more clearly (or maybe I missed it).

13. With respect to comment 12, Fig S8 and line 279. If AlpA levels are 30% lower (line 279) in the *relA/spoT* strain, and the activity of the *alpB-lacZ* fusion is only ~40% lower, maybe the drop in AlpA levels is sufficient to explain at least this effect?

14. Line 366 : perhaps change "the" DNA sequence to "a" DNA sequence.

Reviewer #3 (Remarks to the Author):

This manuscript delves into the function of the regulator AlpA in *Pseudomonas*. AlpA had previously been shown to positively regulate expression of AlpB-E, proteins involved in programmed cell death. Here the authors use RNA-seq and Chip-seq to identify an additional locus that is regulated by AlpA, although the function for this locus remains obscure. Using classical genetics and reporter screens they determine that AlpA acts as an antiterminator by binding to a specific sequence at these promoter regions and by then interacting with the RNAP to enable transcription readthrough. Finally, using transposon mutagenesis, they uncover that the toxic effects upon over-expression of AlpA can be compensated for by disrupting the synthesis of the alarmone (p)ppGpp, which seems to increase the activity of AlpA. This manuscript presents some nicely designed experiments and convincingly demonstrates that AlpA acts as an antiterminator. The intersection between PCD and (p)ppGpp signaling and how this occurs, while beyond the remit of this paper, is definitely of interest to the field and hopefully will be followed up on. I have only minor recommendations/comments.

- For the RNA-seq, please indicate the number of replicates used in the methods.
- From Table S1 it appears that AlpA negatively regulates an operon which includes the regulator BexR. Can the authors comment on this in the discussion. This operon doesn't show up in the Chip-seq analysis, but then neither did the *alpB* promoter due potential competition from AlpR binding. If the Chip-seq had been performed with cells exposed to ciprofloxacin, *alpB* binding may have been apparent. Is it therefore possible that AlpA may bind at a number of other locations on the genome, but this binding might be condition specific?

Following on from this, a binding motif incorporating GGGACGxxxxGGTA was common to both promoters. Have you scanned the genome to look for this motif elsewhere, which might indicate additional condition specific binding sites?

- What mutations were introduced into the -35 and -10 elements of the PA0807 promoter and pAlpB? Perhaps indicate these in Fig 3a.
- Fig 3G – in the methods can you spell out that this is a mix of cells and at what point the cultures were mixed? Presumably after they reached mid log.
- Line 271 – RelA is a synthetase, not a synthase
- Line 275 – in frame mutations of RelA and SpoT were constructed. Can you please specify in the methods or strain table the regions that were deleted i.e just the synthetase domain or was the hydrolase domain included?

- Can you elaborate on the transposon method eg it reads like PA01 was firstly mutagenized and then the plasmid was introduced?
- Over-expression of AlpA is toxic, unless ppGpp levels are lowered. One explanation is the positive effect that ppGpp has on AlpA activity. But it is also possible that over-expression of AlpA somehow increases cellular ppGpp levels, which would be sufficient to shut down growth. Are expression of RelA and/or SpoT altered in the RNA-seq data? Comparing the regulon of over-expressed AlpA to one where ppGpp has been induced from the literature might shed light on this/rule it out.
- If ppGpp is affecting AlpA activity but not AlpA expression (drastically), and binding to the RNAP isn't involved (which is a logical first assumption), then it is likely binding to AlpA itself. This may be outside the remit here, but a simple DRaCALA binding assay with recombinant AlpA and 32P-ppGpp would address this.
- For experiment 6c, how are the E. coli grown? Levels of ppGpp are generally quite low unless the cells are stressed, but here it seems that basal levels of ppGpp are having quite an effect. I would suggest stressing your WT cells with mupirocin for 20 minutes or so to induce the stringent response. Here you should see the opposite phenotype to the deletion mutant, providing more evidence of the link to ppGpp levels.
- Line 422: you mention that the Alp system likely originated from a prophage. Are there AlpA orthologues in other organisms outside the Pseudomonads?

Reviewer #4 (Remarks to the Author):

Summary

The manuscript by Pena and colleagues report on the effect of AlpA as a processive anti-terminator that activates the expression of the alp system, a PCD pathway. Through a comprehensive and elegant series of experiments which include RNA-seq, ChIP-seq, numerous promoter analyses in genetic mutants with ectopic expression of AlpA and AlpR, bacterial two hybrid, the authors identify the additional locus PA0807-0829 to be directly regulated by AlpA, determine the anti-terminator function and target DNA binding sequence of AlpA conserved in the alpB and PA0807 promoters. Interestingly, the AlpA anti-terminator effect is not dependent on the anti-terminator sequence and is positively modulated by ppGpp.

Overall, the manuscript is clearly organized, very well written and a pleasure to read. The authors have recently published on the characterization of AlpA and the alp system of PCD, including the regulation of AlpA of the alp operon. The modulation of AlpA activity by ppGpp is an interesting and novel perspective on the alp system, but the study's results remain preliminary. While the work to demonstrate the function of AlpA as an anti-terminator and to identify its target is well done and of high quality, the impact and novelty of the reported findings appear relatively modest to this reviewer and the work seem best suited for a more targeted microbiology readership.

General comments:

The authors provided comprehensive genetic, transcriptomic and protein binding data to demonstrate the anti-terminator function of AlpA on the alpBCDE gene cluster. These results are clear, rigorous and convincing, with adequate controls provided. The authors provide transcriptomic data to show that PA0807-0829 is also under AlpA control. Although the PA0807 promoter contains the putative terminator location, it would have been a nice addition to provide some experimental data on the AlpA anti-terminator function using the PA0807 promoter.

Several key findings are based on experiments using ectopic expression of AlpA and AlpR. While these constructs are useful in elucidating certain mechanistic interactions difficult to study under

native conditions, they also run the risk of eliciting interactions and processes that are “unphysiological” resulting from protein over-expression. For example, ectopic expression of AlpA results in transcriptional read-through of many genes and bypasses the need for terminator sequences (Fig 5). Could this low lack of specificity be an “artefact” resulting from Alp over-expression? How physiological is this and does it occur when AlpA is induced under more physiological conditions (e.g. in response to DNA damage)? What are the AlpA levels native expression levels compared to those achieved under ectopic expression? Similarly, the authors report that cell growth and viability are reduced upon ectopic expression of AlpR. How does this compare to native AlpR expression under more “physiological” condition such as DNA damage?

Throughout this study, the authors use end point viable counts as a surrogate measure for the PCD phenotype. This is unfortunately a non specific readout that could represent lack of cell growth/replication or any other mechanism of loss of cell viability. While this reviewer recognizes that the cell lysis and PCD phenotype were characterized in a previously published paper, the authors should supplement the viable count data (e.g microscopy and growth curve to show lysis) to better define what low viable cell count represents. In particular, the low viability phenotype associated with pAlpA expression appears to be independent of the alpBDCE lysis genes (Fig 5A). Is this cell lysis? What is it due to? In Fig 3G, the time lapse microscopy shows the reduction in the mCherry+ cell population over time, but it does not show cell lysis per se. The viable cell count data may “mask” several distinct processes and this should be explicitly addressed. The authors also state that “These findings suggest that cells of the Δ relA Δ spoT mutant strain can survive ectopic synthesis of the AlpR-CTD due to a decrease in lysis gene expression, likely as a result of decreased AlpA activity” (line 312). How do the authors reconcile the results in Fig 6B and this conclusion, with the findings that viability is not restored in the alpBCDE mutant + pAlpA (Fig 5A)?

The authors report differential activity of AlpA-responsive genes and bacterial viability in the WT vs relA spot mutant. This observation linking AlpA activity to ppGpp is very interesting but remains relatively superficial. The authors report that site 1 nor site 2 of the RNAP were required for the ppGpp potentiation of AlpA. How ppGpp modulates AlpA activity, whether it is a direct or indirect effect, remains unknown. Can the authors discuss some possibilities? “Through its responsiveness to ppGpp, AlpA enables the integration of environmental cues into the decision to execute a PCD pathway” (Line 351). This conclusion is premature given that the data only compares the WT to a ppGpp-null mutant. It would be more compelling for example to see some dose-dependent effect on the AlpA-PCD pathway in response to varying levels of ppGpp, by inducing native ppGpp accumulation (e.g. amino acid starvation) or through ectopic ppGpp synthesis.

Specific comments:

- The AlpA binding sequence (GGGACG) is conserved in the alpB and PA0807 promoter region. Is it found anywhere else in the PAO1 chromosome?
- The Tn mutagenesis to identify genes required for the toxic effects of ectopically expressed AlpA also identified PA4114 (line 267). Can the authors comment on whether this gene was confirmed to be involved in AlpA toxicity? If yes, what is this gene?
- The ChIP-Seq identified 6 regions. What is the significance of the 5 others? Are these genetic regions under AlpA regulation?
- The authors perform a RNA-seq experiment in the WT vs relA mutant expression pAlpA. As shown in Fig 5A, the WT pAlpA shows minimal viable cells. As such, how reliable are the RNA-seq data if most cells are non viable?
- Line 288: to show that AlpA was more active in the WT compared to the relA mutant, did the authors confirm that the AlpA expression levels were equivalent? The authors have also previously reported that expression of the alpBCDE lysis genes is stochastic and only present in a subset of cells. It would be interesting to determine at the single cell level whether the AlpA expression and its anti-terminator effects are homogenous across the cell population.
- Fig 6C uses an E.coli reporter strain, which presumably expresses a native AlpA. What is the homology of the P.aeruginosa and E.coli AlpA and are the DNA-binding sequences conserved? The authors should expand on a comparison of the two systems. The reduction in alpB reporter activity is 7-fold in the relA spoT E.coli mutant, but only 2-fold in the relA spoT P. aeruginosa strain. Why is that?
- Line 353-365 The role of AlpA-dependent regulation of PA0807-0829 for fitness and virulence

remains highly speculative. "By controlling both the *alpBCDE* and *PA0807-PA0829* operons *AlpA* might provide an efficient means for coordinating both the production and release of factors into the host that facilitate the survival of other *P. aeruginosa* cells." The study has not provided evidence for release of any specific factors, nor their role for *P. aeruginosa* survival within the host.

- Supp table S1, S3 and S4 are missing

Minor comments:

- Fig 2B: the figure shows a statistical difference (19 vs 23) for the *lacZ* expression of the *lacUV5-tR'-tR'* construct. Since there is unlikely any biological significance to this small difference, the statistical significance indicated is irrelevant, and perhaps misleading.

Point-by-point response to reviewer comments

The reviewer comments are in **blue** and our responses are in **black**.

Reviewer #1 (Remarks to the Author):

In this study, the authors analyze transcription regulation by the regulatory protein AlpA from *P. aeruginosa* (PAO1) and conclude that it acts as a processive transcription antiterminator that binds RNAP at the promoter and then travels with the elongation complex and allows read-through of intrinsic transcription terminators. It is shown that deletion of AlpA results in inactivation of transcription from two specific groups of genes under stress conditions, and that AlpA allows terminator read-through in reporter constructs in vivo. Putative promoter and AlpA-responsive elements are further identified in these genes; substitutions of these elements are shown to strongly decrease AlpA effects on reporter gene expression. A two-hybrid screen is then used to identify possible RNAP domains involved in interactions with AlpA. The authors further show that overexpression of AlpA results in dramatic changes in the transcriptome of PAO1, with many read-through transcripts appearing. This suggests that AlpA can act as a non-operon-specific transcription antiterminator in vivo, if expressed at high levels. Finally, it is demonstrated that genes involved in biosynthesis of the alarmone ppGpp (*relA* and *spoT*) modulate the effects of AlpA on transcription and cell survival. However, these effects can not be explained by the effects of ppGpp on RNAP properties, since they are not changed upon deletion of *rpoZ* or *dskA*. Overall, this is an interesting system of a transcription antiterminator in PAO1 that may act similarly to phage-encoded processive antiterminators in *E. coli*. However, many conclusions are based on indirect evidence obtained in vivo and should be confirmed by biochemical studies in vitro with purified proteins, AlpA and RNAP from PAO1. Furthermore, structural analysis of the complexes of AlpA with RNAP and DNA could greatly increase the significance of this work. In the absence of a detailed analysis of the antitermination mechanism, this system remains just another example of transcription regulation, which looks similar to previously analyzed systems – even if it is important for pathogenicity.

Response 1

We agree with the reviewer that it would be interesting to determine the structure of AlpA in combination with RNAP and DNA. However, this is a major and highly specialized endeavor that we believe is well beyond the scope of the current study.

Specific comments:

1) Lines 94-97 – The putative operon activated by AlpA (from PA0807 to PA0830) contains genes in both orientations and only +strand genes are activated. Is anything known about the actual operon structure of this region? Does it contain any additional promoters, or all transcription starts only upstream of 0807? How activation of transcription from the +strand affects transcription of all the genes encoded in the opposite orientation in the –strand?

Response 2

Little is known about the operon structure in this region (from PA0807 to PA0830). The only transcription start sites that have been reported for the +strand are upstream of PA0807 and PA0815 (Gill *et al.* 2018, BMC Genomics 19, 223). AlpA-mediated activation from the +strand doesn't appear to affect transcription of the genes encoded in the opposite orientation in the -strand, at least when AlpA is induced in response to DNA damage (Supplementary Data 1 and new Supplementary Data 2). However, AlpA-mediated activation from the +strand does affect the production of transcripts that are antisense to those genes that are in the opposite orientation (see for example Fig. 1b).

2) Line 114 and below – The ChIP-seq experiments were apparently performed under different conditions from the RNA-seq experiments (in the absence and in the presence of ciprofloxacin) – what are the reasons and whether this could have any effects on the results obtained?

Response 3

As the reviewer notes, the ChIP-Seq experiments with AlpA were performed under conditions that are different from the RNA-Seq experiments. For the ChIP-seq experiments AlpA-TAP was provided from a plasmid to increase the chance we would detect an AlpA enrichment peak. Importantly, the ChIP-seq results allow us to make the case that AlpA targets the *PA0807-PA0829* genes directly and provided the first hint, based on the ChIP-enrichment peak, that AlpA might act as an anti-terminator (Fig. 1b). We do think that performing ChIP-Seq in the absence of DNA damage prevented us from detecting the association between AlpA and the *alpB* promoter. In particular, and as described in the text (lines 122-124), AlpR normally binds the *alpB* promoter in the absence of DNA damage, thus likely preventing AlpA-TAP from binding under these conditions. However, we show that AlpA associates with both the *PA0807* promoter region and the *alpB* promoter region under conditions of DNA damage (i.e. following treatment of cells with ciprofloxacin) by ChIP and qPCR (Supplementary Fig. 2). Under these conditions AlpR undergoes autocleavage (Ref. 12) and thus presumably can no longer prevent AlpA from associating with the *alpB* promoter region.

3) Lines 161-163 – The results of the structural modeling should be included in the figures, if they are discussed in the manuscript.

Response 4

We have included the structural model in *new* Supplementary Fig. 5 (see also Response 18). Note that the structure only includes the putative DNA binding motif.

4) Lines 163-165 – The fact that AlpA is associated with promoter regions (and downstream of them) by itself cannot prove that AlpA is a DNA binding protein, as well as the presence of a predicted DNA-binding structural motif, as well as the presence of a specific DNA motif required for the AlpA action. Although it is plausible that AlpA may indeed interact with DNA in complex with RNAP, direct DNA binding experiments with purified proteins (AlpA and RNAP) could be more conclusive in this case.

Response 5

We agree with the reviewer that DNA binding experiments with purified proteins would be more conclusive. We put considerable effort into trying to purify functional recombinant AlpA from *E. coli* and functional endogenous AlpA from *P. aeruginosa*, but the protein is incredibly insoluble and crashes out of solution almost immediately following purification, making these experiments extremely challenging. We have tried to word the interpretations of our genetic and genomic experiments more carefully to reflect that they support the idea that AlpA is a DNA-binding protein, rather than prove it (lines 179-183 and 236-237).

5) The effects of substitutions at position -14 could have been explained in more detail in the text. As is evident from Supplementary Fig. 5c, nucleotide substitution at position -14 (PalpB - 14*) does not have any effect on promoter activity, while the same substitution in *PA0807* has some additive effects to the substitutions from -20 to -25 (Suppl Fig. 5d). Is this nucleotide proposed to interact with AlpA or RNAP?

Response 6

We think that the nucleotide at this position might contribute to the activity of the *alpB* promoter; note that the basal level of expression of the reporter with the substitution at -14 (with the pEV control) is reduced relative to that of the WT (Supplementary Fig. 6d). However, the basal activity of the *PA0807* promoter does not appear to be influenced by mutation at this position (Supplementary Fig. 6e). Mutation of the *PA0807* promoter at position -14 does have a modest additive effect on AlpA activity when combined with the substitutions from -20 to -25 (Supplementary Fig. 6e), suggesting that position -14 may make a minor contribution to interaction with AlpA, at least at the *PA0807* promoter. We have changed the results section slightly to clarify this (lines 225-226).

6) Lines 221-240 – Two-hybrid experiments suggest that AlpA may interact with the flap domain and region 1.1 of the sigma70 subunit. However, there are several caveats that make their interpretation difficult:

- The experiments were performed in only one ‘orientation’ of the AlpA and RNAP portions in the two-hybrid constructs; reciprocal variants should definitely be tested (in which AlpA is fused to CI and RNAP fragments to alpha CTD).
- The results of a two-hybrid screen cannot be interpreted as a final evidence in support of such interactions, since this method can easily give artefactual signals. While interactions with the flap domain seem plausible (but still should be better confirmed by alternative methods), the interactions with the region sigma 1.1 are doubtful. This region is highly negatively charged and largely unstructured and therefore any DNA binding protein could potentially interact with it. It also remains unexplained in the manuscript why the shorter fragment (1-56) binds better than the longer one (1-93) (see also below)
- The same experiments should be done with *Pseudomonas* constructs (at least for verification of the observed interactions with the flap domain and region 1.1)
- Beta’ 370-417 seems to interact with AlpA (Suppl Fig. 6A, 4-5-fold enhancement) – why it is not included in further analysis?
- For some reasons, the numbers of beta-gal activity (Miller units) are different for the same measurements shown in Fig. 4B and Suppl. Fig. 6.

Response 7

We made two-hybrid constructs with the β flap of RNAP and region 1.1 of σ^{70} fused to the α NTD and linker (effectively replacing the α CTD with the fused fragments). However, the resulting fusions were toxic in our *E. coli* reporter strain, even at low IPTG concentrations, precluding their analysis. The greater toxicity of these RNAP fragments when fused to α likely reflects the fact that the vector supplying the α -fusions is a higher copy number than the one that supplies the λ CI fusion. Moreover, the expression of the α -fusions in our two-hybrid vector is under the control of the strong constitutive *lpp* promoter as well as the IPTG-inducible *lacUV5* promoter, whereas expression of the λ CI fusions, supplied by the other two-hybrid vector, is under the sole control of the *lacUV5* promoter.

We have added additional two-hybrid experiments that show AlpA interacts with region 1.1 of *P. aeruginosa* σ^{70} (new Fig. 4b). Note that we had already shown that AlpA can interact with the β flap of *P. aeruginosa* RNAP (also included in new Fig. 4B). These findings support the idea that AlpA can interact with both the β flap of RNAP and region 1.1 of σ^{70} . Note that we have been careful not to overinterpret our two-hybrid findings and indicate in the Discussion that it remains to be determined whether the interactions between RNAP and AlpA that we detect are important for AlpA activity (lines 422-423 and 449-450).

We decided against pursuing the possible interaction between AlpA and *E. coli* β '(370-417) further, simply because the magnitude of activation in the two-hybrid assay was relatively

modest. In addition, we have not investigated why the shorter fragment of σ^{70} region 1.1 appears to bind AlpA more tightly than the larger fragment. This might be explained by differences in stability of the λ CI-fusions harboring these two fragments, or the effect of the larger fragment on the ability of λ CI to dimerize and thus bind the operator present in the reporter; note that fusions are made to the CTD of λ CI which harbors the dimerization determinant and λ CI must dimerize in order to bind the λ operator. Finally, the two-hybrid assays shown in Supplementary Fig. 7 were performed in a 96-well plate assay whereas those shown in Fig. 4b were performed with cultures grown in glass tubes. In addition, the assays in Fig 4.b were performed with cells that had been grown in the presence of 5 μ M IPTG, whereas those in Supplementary Fig. 7 were performed with cells grown in the presence of 20 μ M IPTG. (We now clarify this in the methods and respective figure legends; line 923 and Supplementary Fig. 7 legend). Differences in the growth conditions and manner in which the relevant absorbance values are read in these assays may account for the observed specific differences in Miller Units between the two figures.

7) The only evidence in support of AlpA interactions with sigma region 1.1 is the two-hybrid screen which is not very conclusive (see above). In the absence of any additional experiments confirming these interactions, any speculations about their possible functional role seem groundless. This interaction may also be difficult to explain from the structural point of view (flap and region 1.1 are expected to be positioned from different sides of the DNA binding cleft). Furthermore, region 1.1 is poorly conserved, and the experiments were performed with the *E. coli* sigma, not *Pseudomonas*.

Response 8

We have added new two-hybrid data that shows AlpA can interact with region 1.1 of *P. aeruginosa* σ^{70} (new Fig. 4b) and respectfully disagree that any speculations about the possible functional role of the interaction we detect between region 1.1 of σ^{70} and AlpA are groundless. Note that we are careful to state in the Discussion that it remains to be determined whether the interaction we detect between region 1.1 of σ^{70} and AlpA plays any role in antitermination (lines 449-450).

8) Lines 290-292 – The RNA-seq profiles for the *relA* strain should be included in Fig. 5B and C, to illustrate what is said in the text here.

Response 9

We have included the RNA-Seq profiles for the *relA* strain in these figures as requested (see new Fig. 5b, c). However, we have replaced Fig. 5c with a region where the difference between the activity of AlpA in the WT and the *relA* mutant strain is closer to 3-fold, to better illustrate our point.

9) Lines 422-423 – It is stated that the Alp system likely originated from a prophage but no evidence in support of this hypothesis is presented in the manuscript. The mechanism of AlpA may indeed be similar to the Q protein of lambdoid phages, but this by itself does not suggest a phage origin (in fact, another antiterminator protein of phage lambda, N appropriated a cellular antitermination complex for its purposes). Does the Alp operon have any features characteristic to prophages, does it contain any know prophage proteins, does it have any differences in the GC content, codon adaptation index etc? What about evolutionary conservation of this system?

Response 10

We have expanded the Discussion slightly to clarify that the *alp* operon does indeed have features characteristic of prophage genes (e.g. *alpB* and *alpC* encode a putative holin and anti-

holin respectively, whose orthologs are typically found in certain prophages) (lines 471-474) (Ref. 12). The system is highly conserved amongst strains of *P. aeruginosa* but doesn't appear to be found outside the pseudomonads (Ref. 12).

10) For all the figures, the explanation of the p-values are missing. There are some numbers of asterisks shown, but they should be clearly explained in each figure. In some cases, the labeling is confusing – for example, in Fig. 2B in the right panel there are four asterisks between 19 and 23 – is this difference significant?

Response 11

We have included the explanation of the p-values to all figure legends and have removed the confusing labelling (see for e.g. revised Fig. 2b).

11) Fig. 1A – There are several genes that are significantly activated by the AlpA deletion – what are these genes? Any explanations?

Response 12

Most of these genes are controlled by a regulator we identified previously called BexR (including the *bexR* gene itself) and are expressed in a bistable fashion (see Supplementary Data 1) (Ref. 16). We have added new RNA-Seq data that suggest these genes are not truly regulated by AlpA. In particular, we have included RNA-Seq data from cells of an *alpA*(stop) mutant treated with ciprofloxacin (new Supplementary Data 2) that we had grown alongside the ciprofloxacin-treated WT and Δ *alpA* mutant cells used in our original RNA-Seq analyses (Supplementary Data 1). Comparing gene expression profiles of WT cells to those of the *alpA*(stop) mutant (which contains a stop codon early on in the *alpA* gene), indicates that AlpA controls the *alpBCDE* genes as well as those in the putative *PA0807* operon, but not *bexR*, or any other BexR-regulated gene. We suspect that the BexR-regulated genes only appear to be upregulated in the *alpA* deletion mutant (Fig. 1a, Supplementary Data 1) because by chance the colony used to inoculate these cultures originated from a cell in which *bexR* is expressed (leading to elevated expression of the entire BexR regulon) whereas the colony used to inoculate the cultures of the WT control originated from a cell in which *bexR* is not expressed. We have added these new data to the results section (lines 100-113) as well as an explanation for why the BexR-regulated genes likely appear in Supplementary Data 1. We have also added a footnote to Supplementary Data 1 to indicate these genes are unlikely to be truly regulated by AlpA (see also Response 33).

12) Fig. 1A – It seems that the ratio of the mutant to the wild-type cells is shown (since the activity of the AlpA-dependent operons is decreased on the plot), while the figure legend says that it is PAO1 to PAO1 alpA.

Response 13

We have fixed this (Fig. 1a legend).

13) Fig. 1B – All the genes from the operon together with their orientation should be clearly labeled in the figure.

Response 14

We have altered Fig 1b in an attempt to make the gene orientations clearer. Specifically, we have elongated the figure and now list the genes encoded on the +strand on a line above those encoded on the -strand.

14) Suppl. Fig. 5C – why the numbers are different from Fig. 3B? (for PalpB Wt and -25 to -20).

Response 15

The numbers for the -25 to -20 constructs are close to one another in the two datasets. Antitermination by AlpA at the PalpB WT reporter is a little higher in Supplementary Fig. 6d (~16-fold) than in Fig. 3b (~11.5 fold). This difference doesn't affect any of our conclusions.

Reviewer #2 (Remarks to the Author):

Peña et al. propose that *Pseudomonas aeruginosa* AlpA positively regulates gene expression by processively antiterminating transcription in a manner analogous to the Q protein of lambdaoid bacteriophages, i.e. it binds to the promoter and travels with RNAP through the operon, preventing termination at intrinsic terminators. The authors identify multiple operons on which AlpA acts, some of which are important for turning on a programmed cell death pathway by activating cell lysis in response to DNA damage. Interestingly, the authors also show that ppGpp enhances antitermination, although the mechanism remains to be worked out. I have a number of questions and minor suggestions the authors may want to consider, but in general, this is a thorough, well-written manuscript that uses ChIP-seq, RNA seq, qPCR, transposase mutagenesis screens, site-directed mutagenesis, and lacZ fusions to identify a heretofore unknown mechanism for regulating gene expression in *Pseudomonas* that integrates environmental responses and virulence. Further biochemical characterization of the mechanism with purified components is probably the next step for determining mechanism. At this point none of the comments below should be considered serious problems.

Response 16

We thank the reviewer for their positive assessment of our manuscript.

Comments

1. Line 155: AlpA's ability to function as an anti-terminator does not appear to be dependent upon specific sequence elements within the terminator itself. Isn't this an overstatement, given that the strength of terminators varies, so the ability of AlpA-modified RNAP to cause readthrough might be dependent on how good the terminator is?

Response 17

We had meant to convey that AlpA's ability to act as an anti-terminator is not dependent upon the specific sequence of the terminator (i.e. that AlpA can promote readthrough of terminators with different sequences). We have altered the wording here for clarification (line 175).

2. Line 160-161: In lieu of an actual structure, adding the PHYRE 2 model as a supplementary figure might be helpful

Response 18

We have included the structural model in *new* Supplementary Fig. 5 (see also Response 4). Note that the structural model only includes the putative DNA-binding motif of AlpA.

3. Lines 169 and following, Fig 5A, etc: The section on the promoter determinants is not as clear as it might be. For example, explain why the absolute activities of the constructs vary, what the actual substitutions are (change from what residue to what other residue, not just the positions), and which mutations define binding sites of AlpA and AlpB. The definition of the minimal region of the PA0807 promoter region that responds to AlpA might deserve its own paragraph.

Probably, there also should be at least one example in which the control and the promoter being tested should be examined using the same terminator.

Response 19

We have attempted to clarify this section and provide possible explanations for why the absolute activities of the constructs vary (lines 188-191). In particular, we note that the reporters used in Supplementary Fig. 6a contain different 5'-untranslated regions upstream of *lacZ* which could result in differences in transcript stability. We also dedicate a new paragraph to the description of the experiments defining the minimal region of the *PA0807* promoter that responds to AlpA (lines 192-208), and (ii) examine the *PA0807* promoter using the same terminator we used in our analysis of the *alpB* promoter (in Fig 2b, i.e. two copies of tR') (*new* Supplementary Fig. 6c).

4. Lines 192-200: The ChIP- qPCR experiment does not really demonstrate that AlpA binds at the position of the mutated sequence (the mutated positions could affect physical contacts nearby). Was any biochemical approach attempted? Does AlpA bind to this region in a footprint or EMSA assay? If no biochemical approaches were attempted to confirm the binding site, then at this point it would be more appropriate to say “putative site”.

Response 20

We attempted EMSA assays with purified AlpA but we suspect the protein used in these experiments was inactive (see also Response 5). We agree with the reviewers point and are now more conservative with our wording—we now refer to this as a putative site as suggested (lines 239-261).

5. Fig. 4: Was a control performed with the same positive-hit λ CI fusions, but without AlpA fused to RNAP to demonstrate that the effect is dependent on an AlpA interaction with λ CI and not an RNAP-RNAP interaction that bypasses the need for AlpA?

Response 21

Yes, in the control experiments in Fig. 4b we show that the same positive hit λ CI fusions with just the α subunit of *E. coli* RNAP being produced from a vector (gray bars) (i.e. without AlpA fused to RNAP). These controls show that the λ CI fusions only interact with AlpA and not with RNAP alone.

6. Line 346 “that AlpA can interact with two distinct portions of RNAP”. Perhaps change “portions of” to “subunits in”.

Response 22

We have made this change (line 392).

7. Line 400: Is it clear what role an interaction between AlpA and region 1.1 of σ^{70} plays in the termination process? Could some of the repressive effects of AlpA observed upon ectopic expression be explained through this interaction? More generally, could AlpA binding reduce transcription by interfering with RPo formation at some promoters when overproduced ectopically?

Response 23

It is not clear what role an interaction between AlpA and region 1.1 of σ^{70} plays in the anti-termination process. (Note that we are careful to state in the Discussion that it remains to be determined whether the interaction we detect between region 1.1 of σ^{70} and AlpA plays any role in anti-termination.) It is certainly possible that the repressive effects of AlpA observed upon

ectopic expression could be explained through this interaction and we allude to that in the Discussion (lines 450-451). Indeed, the only other regulator known to interact with region 1.1 of σ^{70} is Gp2 from bacteriophage T7, which inhibits transcription initiation in *E. coli* by preventing region 1.1 from exiting the active site channel. It could therefore be the case that when ectopically produced, AlpA interferes with the formation of the closed complex. We note in the Discussion that it remains to be determined whether the repressive effects of AlpA we observe are dependent upon its interaction with region 1.1 of σ^{70} (lines 449-450).

8. Line 405: “ppGpp is present in all bacteria”. This should say “almost all bacterial species” (there are a few examples of bacterial species without detectable ppGpp or a *relA* gene).

Response 24

We have made the suggested change (line 453).

9. Fig. S1 : Since the activities in most cases are very low, the colors of the bars in the histograms are often not visible. Perhaps just a label of some sort would work better.

Response 25

We have made changes to the figure for clarity (see *new* Supplementary Fig. 1).

10. Line 259 : perhaps change “in solution” to “off-DNA”?

Response 26

We have made the suggested change (line 302).

11. Lines 321 and 408. This is a semantic issue, but it might help readers understand the mechanism: the lower activity in the absence of ppGpp might be referred to as a reduction in antitermination (or activation) rather than as a reduction in gene expression.

Response 27

We now also refer to this as suggested (lines 365 and 457).

12. With respect to the mechanism by which ppGpp affects AlpB expression, the ability of ppGpp to affect AlpB-lacZ activity in *E. coli* is an important result. Doesn't this mean that a *P. aeruginosa*-specific factor is not required? If neither Site 1 nor Site 2 on RNAP is required, the authors might want to rule out (e.g. by DRaCALA) or at least mention that ppGpp binds to AlpA itself. Otherwise, perhaps the effect is indirect? Maybe this should be stated more clearly (or maybe I missed it).

Response 28

We agree with the reviewer that the ability of ppGpp to affect alpB-lacZ activity in *E. coli* does suggest that a *P. aeruginosa*-specific factor is not required. We also agree that it would be very interesting to test whether AlpA binds ppGpp directly using DRaCALA. However, unfortunately we have been unable to purify active AlpA for these experiments (see also Responses 5 and 41). We have altered the discussion to more clearly infer that ppGpp could potentially act by binding AlpA directly (lines 464-465).

13. With respect to comment 12, Fig S8 and line 279. If AlpA levels are 30% lower (line 279) in the *relA/spoT* strain, and the activity of the alpB-lacZ fusion is only ~40% lower, maybe the drop in AlpA levels is sufficient to explain at least this effect?

Response 29

We think that the apparent small difference between AlpA abundance in cells of the WT and *relA spoT* mutant strains (which does not meet statistical significance; new Supplementary Fig. 8) could account for some of the observed effect (which we now indicate; lines 369-371). We note that in the *relA* mutant strain we see AlpA levels are less than 20% lower than in WT (with the apparent difference not meeting statistical significance) (new Supplementary Fig. 8), but still see that the activity of AlpA appears to be reduced by a factor of 2- to 3-fold in these cells, which can't be explained solely by this small difference.

14. Line 366 : perhaps change “the” DNA sequence to “a” DNA sequence.

Response 30

We have made the suggested change (line 413).

Reviewer #3 (Remarks to the Author):

This manuscript delves into the function of the regulator AlpA in *Pseudomonas*. AlpA had previously been shown to positively regulate expression of AlpB-E, proteins involved in programmed cell death. Here the authors use RNA-seq and Chip-seq to identify an additional locus that is regulated by AlpA, although the function for this locus remains obscure. Using classical genetics and reporter screens they determine that AlpA acts as an antiterminator by binding to a specific sequence at these promoter regions and by then interacting with the RNAP to enable transcription readthrough. Finally, using transposon mutagenesis, they uncover that the toxic effects upon over-expression of AlpA can be compensated for by disrupting the synthesis of the alarmones (p)ppGpp, which seems to increase the activity of AlpA. This manuscript presents some nicely designed experiments and convincingly demonstrates that AlpA acts as an antiterminator. The intersection between PCD and (p)ppGpp signaling and how this occurs, while beyond the remit of this paper, is definitely of interest to the field and hopefully will be followed up on. I have only minor recommendations/comments.

Response 31

We thank the reviewer for their positive assessment of our manuscript.

- For the RNA-seq, please indicate the number of replicates used in the methods.

Response 32

For the RNA-Seq we used 3 biological replicates and include this number in the methods (lines 573 and 586-587).

- From Table S1 it appears that AlpA negatively regulates an operon which includes the regulator BexR. Can the authors comment on this in the discussion. This operon doesn't show up in the Chip-seq analysis, but then neither did the *alpB* promoter due potential competition from AlpR binding. If the Chip-seq had been performed with cells exposed to ciprofloxacin, *alpB* binding may have been apparent. Is it therefore possible that AlpA may bind at a number of other locations on the genome, but this binding might be condition specific?

Response 33

The reviewer is very astute. Most of the genes that appear to be negatively regulated by AlpA in Table S1 are controlled by a regulator we identified previously called BexR (including the *bexR*

gene itself) and are expressed in a bistable fashion (Ref. 16). We have added new RNA-Seq data that suggest these genes are not truly regulated by AlpA. In particular, we have included RNA-Seq data from cells of an *alpA*(stop) mutant treated with ciprofloxacin (*new* Supplementary Data 2) that we had grown alongside the ciprofloxacin-treated WT and Δ *alpA* mutant cells used in our original RNA-Seq analyses (Supplementary Data 1). Comparing gene expression profiles of WT cells to those of the *alpA*(stop) mutant (which contains a stop codon early on in the *alpA* gene), indicates that AlpA controls the *alpBCDE* genes as well as those in the putative *PA0807* operon, but not *bexR*, or any other BexR-regulated gene. We suspect that the BexR-regulated genes only appear to be upregulated in the *alpA* deletion mutant (Supplementary Data 1) because by chance the colony used to inoculate these cultures originated from a cell in which *bexR* is expressed (leading to elevated expression of the entire BexR regulon) whereas the colony used to inoculate the cultures of the WT control originated from a cell in which *bexR* is not expressed. We have added these new data to the results section (lines 100-113) as well as an explanation for why the BexR-regulated genes likely appear in Supplementary Data 1. We have also added a footnote to Supplementary Data 1 to indicate these genes are unlikely to be truly regulated by AlpA. Note also that none of the other possible AlpA binding sites (see Response 34 below) are associated with *bexR* (*PA2432*), or any other BexR-regulated gene in our dataset. Because we don't think *bexR* or the other BexR-regulated genes are true targets of AlpA we do not mention them in the Discussion (see also Response 12).

Following on from this, a binding motif incorporating GGGACGxxxxGGTA was common to both promoters. Have you scanned the genome to look for this motif elsewhere, which might indicate additional condition specific binding sites?

Response 34

We have scanned the genome and find the binding motif GGGACGxxxxxGGTA a total of seven times. Two of these correspond to those found at the *PA0807* and *alpB* promoters, four are within genes (*PA2042*, *PA2711*, *PA2829*, and *PA3083*), and one is upstream of a gene (*PA3249*), but on the opposite strand.

- What mutations were introduced into the -35 and -10 elements of the *PA0807* promoter and *pAlpB*? Perhaps indicate these in Fig 3a.

Response 35

We have indicated these in revised Supplementary Fig. 3 that describes the effects of these mutations (see Supplementary Fig. 3a).

- Fig 3G – in the methods can you spell out that this is a mix of cells and at what point the cultures were mixed? Presumably after they reached mid log.

Response 36

We now spell out in the methods that cells were grown separately to the mid-log phase, treated with IPTG for 30 minutes, then mixed just prior to placing them on an agar pad for microscopy (lines 503 and 677).

- Line 271 – *RelA* is a synthetase, not a synthase

Response 37

We have corrected this (line 314).

- Line 275 – in frame mutations of *RelA* and *SpoT* were constructed. Can you please specify in

the methods or strain table the regions that were deleted i.e just the synthetase domain or was the hydrolase domain included?

Response 38

For both the $\Delta reIA$ and $\Delta spoT$ mutations, the in-frame deletions contained the original start and stop codons of the respective gene separated by DNA specifying three alanine codons. Thus, essentially the entire gene was deleted in each case. We now include this information in the methods section (lines 527-529).

- Can you elaborate on the transposon method eg it reads like PA01 was firstly mutagenized and then the plasmid was introduced?

Response 39

We have elaborated on the transposon method to make it clearer that PA01 was first mutagenized with the transposon and then the plasmid encoding AlpA was introduced (lines 691-692).

- Over-expression of AlpA is toxic, unless ppGpp levels are lowered. One explanation is the positive effect that ppGpp has on AlpA activity. But it is also possible that over-expression of AlpA somehow increases cellular ppGpp levels, which would be sufficient to shut down growth. Are expression of RelA and/or SpoT altered in the RNA-seq data? Comparing the regulon of over-expressed AlpA to one where ppGpp has been induced from the literature might shed light on this/rule it out.

Response 40

The reviewer raises an interesting possibility. However, the RNA-Seq studies suggest that ectopic expression of AlpA does not appreciably alter the expression of *spoT* (PA5338) and actually reduces the expression of *relA* (PA0934) by a factor of 2 (see Supplementary Data 3).

- If ppGpp is affecting AlpA activity but not AlpA expression (drastically), and binding to the RNAP isn't involved (which is a logical first assumption), then it is likely binding to AlpA itself. This may be outside the remit here, but a simple DRaCALA binding assay with recombinant AlpA and ³²P-ppGpp would address this.

Response 41

This is a great suggestion. Unfortunately, we have been unable to purify active recombinant AlpA for these experiments (see also Responses 5 and 28).

- For experiment 6c, how are the *E. coli* grown? Levels of ppGpp are generally quite low unless the cells are stressed, but here it seems that basal levels of ppGpp are having quite an effect. I would suggest stressing your WT cells with mupirocin for 20 minutes or so to induce the stringent response. Here you should see the opposite phenotype to the deletion mutant, providing more evidence of the link to ppGpp levels.

Response 42

For this experiment cells of *E. coli* are grown to mid-log in LB, conditions where the levels of ppGpp would indeed be expected to be generally low. We did try an experiment analogous to the one suggested where we induced the stringent response using serine hydroxamate in wild-type cells of our reporter strain but didn't observe any dramatic increase in β -galactosidase activity. This is somewhat difficult to interpret as the conditions we used have been reported to result in such a large and rapid increase in ppGpp production that protein synthesis is rapidly

inhibited. What we would like to be able to do is increase ppGpp production in small increments, such that protein synthesis is not completely inhibited, and then test what effect this has on expression of the AlpA-controlled reporter. However, we don't currently have such a system up and running in the lab, and the development of such a system would be a significant undertaking that we feel is outside the scope of the current study.

• Line 422: you mention that the Alp system likely originated from a prophage. Are there AlpA orthologues in other organisms outside the Pseudomonads?

Response 43

We do not find AlpA orthologues in organisms outside the Pseudomonads (see also Response 10).

Reviewer #4 (Remarks to the Author):

Summary

The manuscript by Pena and colleagues report on the effect of AlpA as a processive anti-terminator that activates the expression of the alp system, a PCD pathway. Through a comprehensive and elegant series of experiments which include RNA-seq, ChIP-seq, numerous promoter analyses in genetic mutants with ectopic expression of AlpA and AlpR, bacterial two hybrid, the authors identify the additional locus PA0807-0829 to be directly regulated by AlpA, determine the anti-terminator function and target DNA binding sequence of AlpA conserved in the alpB and PA0807 promoters. Interestingly, the AlpA anti-terminator effect is not dependent on the anti-terminator sequence and is positively modulated by ppGpp.

Overall, the manuscript is clearly organized, very well written and a pleasure to read. The authors have recently published on the characterization of AlpA and the alp system of PCD, including the regulation of AlpA of the alp operon. The modulation of AlpA activity by ppGpp is an interesting and novel perspective on the alp system, but the study's results remain preliminary. While the work to demonstrate the function of AlpA as an anti-terminator and to identify its target is well done and of high quality, the impact and novelty of the reported findings appear relatively modest to this reviewer and the work seem best suited for a more targeted microbiology readership.

General comments:

The authors provided comprehensive genetic, transcriptomic and protein binding data to demonstrate the anti-terminator function of AlpA on the alpBCDE gene cluster. These results are clear, rigorous and convincing, with adequate controls provided. The authors provide transcriptomic data to show that PA0807-0829 is also under AlpA control. Although the PA0807 promoter contains the putative terminator location, it would have been a nice addition to provide some experimental data on the AlpA anti-terminator function using the PA0807 promoter.

Response 44

We thank the reviewer for their comments. We have added an experiment that shows AlpA can act as an anti-terminator using a minimal *PA0807* promoter positioned upstream of tandem tR' terminators (*new* Supplementary Fig. 6c). We have also added an experiment that supports the existence of a terminator downstream of the *PA0807* promoter (*new* Supplementary Fig. 4b).

Several key findings are based on experiments using ectopic expression of AlpA and AlpR. While these constructs are useful in elucidating certain mechanistic interactions difficult to study under native conditions, they also run the risk of eliciting interactions and processes that are “unphysiological” resulting from protein over-expression. For example, ectopic expression of AlpA results in transcriptional read-through of many genes and bypasses the need for terminator sequences (Fig 5). Could this low lack of specificity be an “artefact” resulting from Alp over-expression? How physiological is this and does it occur when AlpA is induced under more physiological conditions (e.g. in response to DNA damage)? What are the AlpA levels native expression levels compared to those achieved under ectopic expression? Similarly, the authors report that cell growth and viability are reduced upon ectopic expression of AlpR. How does this compare to native AlpR expression under more “physiological” condition such as DNA damage?

Response 45

The reviewer makes an excellent point about ectopic production of AlpA. We believe that the toxic effects of AlpA we observe upon overproduction of AlpA may not be physiologically relevant. As the reviewer points out, ectopic production of AlpA appears to result in the transcriptional readthrough of many genes. However, when we look at which genes are regulated by AlpA in response to DNA damage, we observe essentially only the *alpBCDE* self-lysis genes and genes in the *PA0807-PA0829* putative operon being controlled, at least at the time-point we analyzed following DNA damage (Fig. 1a, Supplementary Data 1 and 2). We infer from this that the widespread readthrough of transcription terminators we observe following overproduction of AlpA is because we are producing the protein at artificially high levels. Indeed, our RNA-Seq studies suggest that *alpA* transcript abundance is at least 10-times higher than basal in cells following DNA damage (Supplementary Data 1), but ~270 times higher than basal when we induce from a plasmid (Supplementary Data 3). However, DNA damage induces *alpA* expression only in a subset of cells and we can't rule out the possibility that at time points following DNA damage later than that used here, more widespread readthrough of terminators might occur. Regardless, our findings with ectopic production of AlpA are important as they lead us to identify a role for ppGpp in modulating the activity of AlpA.

With respect to ectopic production of AlpR, we only ectopically produce the C-terminal domain of AlpR (the AlpR-CTD) in cells that contain native amounts of full-length AlpR. This sequesters native full-length AlpR into inactive heterodimers of AlpR:AlpR-CTD that can no longer bind the DNA (Ref. 12). This therefore mimics conditions of DNA damage but doesn't directly alter the abundance of AlpR. Under these conditions, induction of the *alp* system results in PCD in a manner that is entirely dependent upon the *alpBCDE* genes (Ref. 12; see also Fig. 6b). We therefore think that this more closely mimics what happens in response to DNA damage (as in Fig. 1a, b).

Throughout this study, the authors use end point viable counts as a surrogate measure for the PCD phenotype. This is unfortunately a non specific readout that could represent lack of cell growth/replication or any other mechanism of loss of cell viability. While this reviewer recognizes that the cell lysis and PCD phenotype were characterized in a previously published paper, the authors should supplement the viable count data (e.g microscopy and growth curve to show lysis) to better define what low viable cell count represents. In particular, the low viability phenotype associated with pAlpA expression appears to be independent of the *alpBCDE* lysis genes (Fig 5A). Is this cell lysis? What is it due to? In Fig 3G, the time lapse microscopy shows the reduction in the mCherry+ cell population over time, but it does not show cell lysis per se. The viable cell count data may “mask” several distinct processes and this should be explicitly

addressed. The authors also state that “These findings suggest that cells of the Δ relA Δ spoT mutant strain can survive ectopic synthesis of the AlpR-CTD due to a decrease in lysis gene expression, likely as a result of decreased AlpA activity” (line 312). How do the authors reconcile the results in Fig 6B and this conclusion, with the findings that viability is not restored in the *alpBCDE* mutant + pAlpA (Fig 5A)?

Response 46

We have added a video of the time-lapse microscopy we performed in Fig. 3g which demonstrates PCD occurs through cell lysis under these conditions (see *new* Supplementary Movie 1). We have not added additional time-lapse microscopy images to the studies in which we overproduce AlpA as we believe we aren’t making physiologically relevant amounts of AlpA here (see Response 45 above). In fact, our results indicate that ectopic overproduction of AlpA is toxic, resulting either in inhibition of cell growth or in cell death regardless of whether or not the *alpBCDE* genes are present (Fig 5a). We have not addressed which of the many genes whose expression alter upon ectopic production of AlpA might (Supplementary Data 3) be responsible for this observed effect on growth or cell viability that occurs even in the absence of the *alpBCDE* cell lysis genes.

For experiments involving the AlpR-CTD (as in Fig. 6b) we are not producing AlpA at artificially high levels (see Response 45). Under these conditions AlpA is induced (through sequestration of endogenous AlpR) to a level that is sufficient to result in PCD through induction of *alpBCDE*. Thus, cells lacking *alpBCDE* completely survive induction of the AlpR-CTD. In Fig. 5a, as noted above, we are producing AlpA at artificially high levels that are toxic, regardless of whether or not *alpBCDE* are present.

The authors report differential activity of AlpA-responsive genes and bacterial viability in the WT vs *relA* spot mutant. This observation linking AlpA activity to ppGpp is very interesting but remains relatively superficial. The authors report that site 1 nor site 2 of the RNAP were required for the ppGpp potentiation of AlpA. How ppGpp modulates AlpA activity, whether it is a direct or indirect effect, remains unknown. Can the authors discuss some possibilities? “Through its responsiveness to ppGpp, AlpA enables the integration of environmental cues into the decision to execute a PCD pathway” (Line 351). This conclusion is premature given that the data only compares the WT to a ppGpp-null mutant. It would be more compelling for example to see some dose-dependent effect on the AlpA-PCD pathway in response to varying levels of ppGpp, by inducing native ppGpp accumulation (e.g. amino acid starvation) or through ectopic ppGpp synthesis.

Response 47

We have expanded the Discussion slightly to include possibilities for how the binding of ppGpp directly to AlpA might promote its activity (i.e. by modulating the interaction of AlpA with the DNA or with RNA polymerase) (lines 464-465). We have also toned down our conclusion concerning ppGpp (line 397). We agree with the reviewer that it would be interesting to see a dose-dependent effect of ppGpp on the AlpA-PCD pathway. Ideally, we would like to be able to increase ppGpp production in small increments, such that protein synthesis is not completely inhibited, and then test what effect this has on expression of the AlpA-controlled reporter. However, we don’t currently have such a system up and running in the lab (see also Response 42).

Specific comments:

- The AlpA binding sequence (GGGACG) is conserved in the *alpB* and PA0807 promoter

region. Is it found anywhere else in the PAO1 chromosome?

Response 48

The *P. aeruginosa* genome is very GC-rich so this sequence is found frequently (a total of 2,805 times). However, the sequence that AlpA recognizes may be more expansive than this (see Response 34).

- The Tn mutagenesis to identify genes required for the toxic effects of ectopically expressed AlpA also identified PA4114 (line 267). Can the authors comment on whether this gene was confirmed to be involved in AlpA toxicity? If yes, what is this gene?

Response 49

We did not follow up on whether PA4114 was required for the toxic effects of AlpA.

- The ChIP-Seq identified 6 regions. What is the significance of the 5 others? Are these genetic regions under AlpA regulation?

Response 50

We don't think the 5 other regions we identified as associated with AlpA are subject to AlpA control under the conditions of our experiments (i.e. when cells are grown in LB); we don't observe any change in gene expression in these regions when the *alp* system is induced in cells in response to DNA damage (Fig. 1a; Supplementary Data 1 and 2).

- The authors perform a RNA-seq experiment in the WT vs *relA* mutant expression pAlpA. As shown in Fig 5A, the WT pAlpA shows minimal viable cells. As such, how reliable are the RNA-seq data if most cells are non viable?

Response 51

The experiments in Fig 5a that show minimal cell viability upon ectopic synthesis of AlpA are end-point assays where cells are incubated on plates overnight in the presence or absence of IPTG [1 mM]. Under these conditions, IPTG induces the synthesis of AlpA resulting in inhibition of cell growth or resulting in cell death. The RNA-seq experiments comparing the effects of ectopic AlpA production in wild-type and $\Delta relA$ mutant cells are performed differently, with cells being first grown in LB liquid for 2 hours to an OD₆₀₀ of ~0.04. IPTG was then added to induce synthesis of AlpA (at a final concentration of 1 mM) and cells were harvested for RNA isolation 90 minutes later. Under these conditions, the majority of cells are still viable (as revealed by plating) and the OD₆₀₀ of the cultures used for RNA isolation (for RNA-Seq) are comparable. In particular, the OD₆₀₀ of the triplicate cultures used for RNA isolation were as follows: WT with pAlpA (0.36-0.38), WT with pEV (0.36-0.42), $\Delta relA$ with pAlpA (0.3-0.34), and $\Delta relA$ with pEV (0.3-0.4). We have added this latter information to the Methods section for clarity.

- Line 288: to show that AlpA was more active in the WT compared to the *relA* mutant, did the authors confirm that the AlpA expression levels were equivalent? The authors have also previously reported that expression of the *alpBCDE* lysis genes is stochastic and only present in a subset of cells. It would be interesting to determine at the single cell level whether the AlpA expression and its anti-terminator effects are homogenous across the cell population.

Response 52

We have quantified AlpA-V abundance in wild-type and *relA* mutant cells and see that AlpA abundance is reduced by approximately 20% in these cells when compared to WT (*new* Supplementary Fig. 8), a difference that does not meet statistical significance. We agree with

the reviewer that it would be interesting to test whether *alpA*, like the *alpBCDE* lysis genes it controls, is made in cells in a stochastic fashion in response to DNA damage. However, we believe the proposed experiments would be tangential to the current study.

- Fig 6C uses an *E.coli* reporter strain, which presumably expresses a native AlpA. What is the homology of the *P.aeruginosa* and *E.coli* AlpA and are the DNA-binding sequences conserved? The authors should expand on a comparison of the two systems. The reduction in *alpB* reporter activity is 7-fold in the *relA spoT E.coli* mutant, but only 2-fold in the *relA spoT P. aeruginosa* strain. Why is that?

Response 53

In Fig. 6c we are simply producing *P. aeruginosa* AlpA in *E. coli*. We don't know why the effects of the *relA spoT* mutations on *alpB* reporter activity are more pronounced in *E. coli* than in *P. aeruginosa*. It is conceivable it could be explained by differences in the abundance of the basal amount of ppGpp being made. However, additional experiments would need to be performed to test this specific possibility.

- Line 353-365 The role of AlpA-dependent regulation of PA0807-0829 for fitness and virulence remains highly speculative. "By controlling both the *alpBCDE* and PA0807-PA0829 operons AlpA might provide an efficient means for coordinating both the production and release of factors into the host that facilitate the survival of other *P. aeruginosa* cells." The study has not provided evidence for release of any specific factors, nor their role for *P. aeruginosa* survival within the host.

Response 54

We agree with the reviewer that the AlpA-dependent control of PA0807-0829 for virulence remains speculative. However, there are published data (that we cite; Ref. 13) that PA0807 is important for virulence in an acute lung infection model in mice. Moreover, our prior studies suggest that the *alp* genes are expressed in a subset of cells in an acute lung infection model in mice (Ref. 12). It is therefore possible that PA0807 is expressed in an AlpA-dependent fashion in certain cells in the mouse lung infection model. Finally, our prior studies suggest that the AlpA system is induced in a subset of cells in response to DNA damage, and that when induced in cells, those cells lyse in an *alpBCDE*-dependent fashion (Ref. 12). Cell lysis results in the release of cellular contents. We therefore envision that induction of the Alp system in a subset of cells could release the contents of these cells producing the products of the PA0807-PA0829 operon, which could then promote the survival of the remainder of the cell population. Nevertheless, we have reworded this sentence slightly to more clearly highlight the speculative nature of our suggestion (line 409).

- Supp table S1, S3 and S4 are missing

Response 55

Actually, these Tables were included in the original submission. In the resubmission they are included as Supplementary Data 1, 3, and 4.

Minor comments:

- Fig 2B: the figure shows a statistical difference (19 vs 23) for the *lacZ* expression of the *lacUV5 – tR'-tR'* construct. Since there is unlikely any biological significance to this small difference, the statistical significance indicated is irrelevant, and perhaps misleading.

Response 56

We agree with the reviewer that these very small differences in *lacZ* expression are unlikely to be biologically significant and thus irrelevant. We have removed the statistical comparison (see revised Fig. 2b).

REVIEWER COMMENTS

Reviewer #1 (Remarks to the Author):

In the revised manuscript, the authors have responded to many (but not all) comments from the original reviews and provided some additional data. However, there are still serious questions remaining about the putative mechanism of transcription antitermination by AlpA, and some experiments still require more careful explanation/interpretation. Importantly, the manuscript describes a new example of a cellular transcription antiterminator protein that is important for virulence in *P. aeruginosa*. Similarly to the Q protein of phage lambda, AlpA is proposed to bind RNAP at promoters and travel with it along the operon sequence, thus allowing transcription of downstream genes. However, the molecular details of this process are missing, and it remains to be shown whether AlpA indeed interacts with its putative binding site within promoter complexes and then loads onto RNAP. How it protects the transcription complex from termination is also unknown (however, given its broad effects on antisense transcription during overexpression, it should probably act on both intrinsic and rho-dependent terminators). Furthermore, the role of ppGpp in antitermination remains enigmatic. Thus, from the mechanistic point of view, the manuscript adds relatively little to our understanding of various pathways of transcription regulation.

Specific comments:

1) The chromosomal locus containing genes PA0807-PA0829 is still called an 'operon' in several places, while it is highly unlikely that this region is normally transcribed as a single mRNA. It contains many genes oriented in the opposite direction, and such organization of operons is highly unusual. Furthermore, as the authors indicate in their response, there is at least one additional positive strand promoter before gene PA0815. Is it possible that the natural target of AlpA is only the first gene, while increased transcription of the remaining genes is a 'noise' resulting from the activation of the upstream transcription? Later in the manuscript, the authors demonstrate that AlpA can activate many noncoding transcripts, possibly because of its nonspecific antitermination effects on the whole-genome level. Is it possible that a similar effect is observed here for this particular locus? Is there any evidence that the whole region is transcribed as a single RNA transcript? Are these genes functionally connected? If no, they can be better called simply 'locus', not 'operon'.

2) I still suggest to downplay the statement that AlpA interacts with sigma region 1.1, and that these interactions may be important for regulation. The only evidence for this is found in two-hybrid data, and it is difficult to imagine how such interactions can occur in the context of an open promoter complex, together with the beta flap interactions (which are looking more plausible because many antiterminators bind in the same place). It is also unlikely that such interactions can persist in the elongation complex – this would be an exciting possibility but without further biochemical evidence the proposed interaction looks doubtful. For example, the observed two-hybrid signal may result from non-specific interactions with the negatively charged N-terminal region of the sigma factor. Another caveat is the absence of reciprocal controls in the two hybrid screen, which could not be performed for technical reasons.

3) The mechanism behind for the observed effect of the relA deletion on the AlpA-mediated antitermination is still unclear, and the effect itself is in fact relatively small (2-3 fold at most). From this point of view, it is very important to be sure that the effects of the relA deletion on activation of AlpA-dependent transcripts is indeed seen only below terminators (at least for known terminator sequences). From Fig. 5 b and c, it is no clear whether the upstream operon parts before terminators are less affected by the relA deletion than the downstream parts (because the levels of transcription in the wt strain are higher than the y-scale and the peaks are not completely visible). Some quantification is definitely warranted, to be sure that the relA deletion does not simply decrease transcription of the whole operons or antisense RNAs. Supplemental data 3 and 4 show the list of genes that are differentially expressed depending on the AlpA expression in the wt and relA strains – however, since the lists of these genes are different for the two strains, their direct comparison is complicated. In particular, it would be good to compare the numbers (fold-changes in gene expression) for the same list of genes in both strains in parallel, including the first genes before the terminator. Furthermore, even 30% changes in the level of AlpA expression in

the *relA* strain may translate to much larger changes in the expression of its target transcripts, depending on its actual affinity to RNAP and its free concentration in the cell. Importantly, the observed changes in the levels of expression of the target genes are not 1000-fold (this large change is in cell titers, which non-linearly depend on changes in gene expression), but much smaller (only 2-fold for *alpD*, for example, line 346). Furthermore, given the fact that neither deletion of the omega subunit nor deletion of both *DksA* factors change antitermination by AlpA, the actual mechanism of ppGpp-dependent activation remains obscure, and it may be either direct or indirect. So, I suggest to remove the statement that AlpA is a ppGpp-responsive antiterminator, until the mechanism explaining the observed effects is clear.

4) Overexpression of TAP-tagged AlpA reveals its binding to only 6 chromosomal regions (Fig.1); however, in another experiments overexpression of AlpA results in significant changes in the expression of hundreds of genes and antisense transcripts (most of which are activated). Are the levels of AlpA expression comparable in these two cases? Even if activation of transcription occurs through nonspecific binding of AlpA to the elongation complex, is not it expected that AlpA should be enriched in these regions in the ChIP experiments? For the remaining five peaks in the ChIP experiments, can a putative binding site for AlpA be found within the target region? Is transcription of these loci activated by AlpA?

5) Deletions of *DksA*/*DksA2* in fact increase the activity of reporter constructs in response to expression of pAlpA (~2-fold) (Fig. S10b). Do the authors have any explanation? So, is it possible to say that *DksA* counteracts the effects of AlpA on termination, e.g. by promoting termination?

6) The schemes of DNA constructs containing promoter fragments could be added to Fig. S6, to make it more clear.

Reviewer #2 (Remarks to the Author):

The authors have patiently and thoroughly responded to the 56 comments made by the 4 referees point-by-point. Although the absence of an in vitro system with purified AlpA makes some conclusions speculative, the authors have appropriately toned down these conclusions in response to the reviewers' comments. I have no further comments.

Reviewer #3 (Remarks to the Author):

All of my comments have been addressed. I think this is a clear and well executed paper and deserves publication.

to note: the colouring and layout of the DNA sequences in Fig 3a has gotten messed up in the merged PDF - something to check

Reviewer #4 (Remarks to the Author):

The authors have provided a revised and improved manuscript and included additional data to response to previous comments. The revisions provide greater clarity and new evidence to address many comments from myself and the other reviewers. However, I still have several concerns outlined below.

Response 45. I remain concerned about the interpretation of results generated with the ectopic AlpA expression which is ~27X higher than those levels achieved with DNA damage and 270X higher than basal levels according to the authors. The authors confirm that this overexpression is not physiological and that the transcriptional readthrough of many genes likely does not occur in response to DNA damage. In my opinion, there is thus no direct evidence that the bypassing of

termination sequences (Fig 5b) occurs under physiological conditions (such as DNA damage), and if it doesn't, what is the biological relevance of the observed results, which only occur as a result of excessively high AlpA expression? This experimental system and the evidence presented thus fail to convince me that those results (Fig 5b) provide meaningful biological insight.

Response 46 The authors have included a time-lapse microscopy which support the observed cell lysis phenotype. With regards to the toxicity of AlpA ectopic expression, the authors state that the low viability encountered with AlpA ectopic production may not be related to cell lysis per se and toxicity could be due to various mechanisms, potentially unrelated to PCD. In the absence of a specific mechanism of toxicity or interaction to explain the modulatory effect of ppGpp, the observation that ppGpp modulates the activity of AlpA is of relatively modest mechanistic significance since ppGpp has such pleiotropic effects.

Minor comments:

Response 49: The Tn mutagenesis was performed in order to determine the genes involved in the toxic effects of AlpA ectopic expression. It would be informative to explicitly mention that the role of PA4114 was not further investigated.

Response 50: Line 117 The authors state that "These ChIP-Seq analyses revealed that AlpA associates with 6 regions of the PAO1 chromosome... Fig 1b and Supp table 1". The Fig 1b report on the region PA0807-0830 and the Supp Table 1 is the RNA seq data for alpA mutant vs WT with cipro. The authors should report what the other 5 regions are. Since the authors do not think that these 5 other regions are relevant, this should be stated explicitly.

Response 48 The putative AlpA binding sequence should be explicitly stated in the text or figure legend for clarity, as it is only shown in Fig 3. Furthermore, the author should also report where this binding motif is found in the genome (as per Response 34) and discuss what those results mean since the other genes are not identified in the RNA-seq or ChIP-Seq analyses to be AlpA regulated.

Point-by-point response to reviewer comments

The reviewer comments are in **blue** and our responses are in **black**.

Reviewer #1 (Remarks to the Author):

In the revised manuscript, the authors have responded to many (but not all) comments from the original reviews and provided some additional data. However, there are still serious questions remaining about the putative mechanism of transcription antitermination by AlpA, and some experiments still require more careful explanation/interpretation. Importantly, the manuscript describes a new example of a cellular transcription antiterminator protein that is important for virulence in *P. aeruginosa*. Similarly to the Q protein of phage lambda, AlpA is proposed to bind RNAP at promoters and travel with it along the operon sequence, thus allowing transcription of downstream genes. However, the molecular details of this process are missing, and it remains to be shown whether AlpA indeed interacts with its putative binding site within promoter complexes and then loads onto RNAP. How it protects the transcription complex from termination is also unknown (however, given its broad effects on antisense transcription during overexpression, it should probably act on both intrinsic and rho-dependent terminators). Furthermore, the role of ppGpp in antitermination remains enigmatic. Thus, from the mechanistic point of view, the manuscript adds relatively little to our understanding of various pathways of transcription regulation.

Response 1

We appreciate that there is still much to learn about the mechanism by which this newly described transcription antiterminator acts. However, we believe our study is important as AlpA represents the first processive antiterminator to play a role in the control of virulence gene expression in *Pseudomonas aeruginosa*. Moreover, we identify for the first time a second set of genes that AlpA controls, define a sequence that is important for AlpA activity and association at target promoters, use a two-hybrid assay to identify portions of RNA polymerase that AlpA can interact with and present evidence that the activity of AlpA may be modulated by ppGpp.

Specific comments:

1) The chromosomal locus containing genes PA0807-PA0829 is still called an 'operon' in several places, while it is highly unlikely that this region is normally transcribed as a single mRNA. It contains many genes oriented in the opposite direction, and such organization of operons is highly unusual. Furthermore, as the authors indicate in their response, there is at least one additional positive strand promoter before gene PA0815. Is it possible that the natural target of AlpA is only the first gene, while increased transcription of the remaining genes is a 'noise' resulting from the activation of the upstream transcription? Later in the manuscript, the authors demonstrate that AlpA can activate many noncoding transcripts, possibly because of its nonspecific antitermination effects on the whole-genome level. Is it possible that a similar effect is observed here for this particular locus? Is there any evidence that the whole region is transcribed as a single RNA transcript?

Are these genes functionally connected? If no, they can be better called simply 'locus', not 'operon'.

Response 2

We respectfully disagree with the Reviewer and think it quite likely that the PA0807-PA0829 locus represents an operon, albeit an unusual one. Operons in *E. coli* and *P. aeruginosa* are thought to be as large as 25 Kb, suggesting that RNAP is capable of transcribing long stretches of DNA without terminating. We do not have any direct evidence that a single transcript can extend from PA0807-PA0831. However, ectopic expression of *alpA* results in a large increase in

expression of this putative operon (from *PA0807-PA0831*) (Supplementary Data 3), which is consistent with this possibility and argues against the alternative idea proposed by the reviewer that the first gene is the only target. For example, ectopic expression of *alpA* results in (i) a 110-fold increase in expression of *PA0808* (the second gene in the putative operon), (ii) a 103-fold increase in expression of antisense *PA0813* (upstream of *PA0815*), and (iii) a 69-fold increase in expression of antisense *PA0826* (far downstream of *PA0815*) (Supplementary Date 3). Nevertheless, we have altered the text to refer to this region either as a putative operon or locus.

2) I still suggest to downplay the statement that AlpA interacts with sigma region 1.1, and that these interactions may be important for regulation. The only evidence for this is found in two-hybrid data, and it is difficult to imagine how such interactions can occur in the context of an open promoter complex, together with the beta flap interactions (which are looking more plausible because many antiterminators bind in the same place). It is also unlikely that such interactions can persist in the elongation complex – this would be an exciting possibility but without further biochemical evidence the proposed interaction looks doubtful. For example, the observed two-hybrid signal may result from non-specific interactions with the negatively charged N-terminal region of the sigma factor. Another caveat is the absence of reciprocal controls in the two hybrid screen, which could not be performed for technical reasons.

Response 3

We are reluctant to ignore the possibility entirely that the interaction we detect between region 1.1 of σ^{70} and AlpA may be important for regulation. We note that the location and structure of region 1.1 of σ^{70} in an open promoter complex has not been clearly defined and recent evidence from single molecule studies suggests that a significant subpopulation of transcription elongation complexes retain σ^{70} throughout elongation. Nevertheless, we have attempted to downplay this statement even further. In the Results section we state that “The results of our two-hybrid assays *raise the possibility* that AlpA may exert its regulatory effects through interaction with either the β -flap and/or region 1.1 of σ^{70} ” (lines 279-280). In addition, in the Discussion we now state that (i) it is unclear how an interaction between AlpA and region 1.1 of σ^{70} could influence termination (lines 444-445), (ii) that we cannot rule out the possibility that the interaction we detect between AlpA and region 1.1 of σ^{70} in our two-hybrid assay is due simply to the fact that this region of σ^{70} is negatively charged (lines 445-447), and (iii) that “Further work will be required to determine whether an interaction with region 1.1 of σ^{70} plays any role in AlpA-mediated antitermination and whether any of the observed repressive effects of AlpA might be explained through this *potential* interaction” (lines 453-456).

3) The mechanism behind for the observed effect of the *relA* deletion on the AlpA-mediated antitermination is still unclear, and the effect itself is in fact relatively small (2-3 fold at most). From this point of view, it is very important to be sure that the effects of the *relA* deletion on activation of AlpA-dependent transcripts is indeed seen only below terminators (at least for known terminator sequences). From Fig. 5 b and c, it is no clear whether the upstream operon parts before terminators are less affected by the *relA* deletion than the downstream parts (because the levels of transcription in the wt strain are higher than the y-scale and the peaks are not completely visible). Some quantification is definitely warranted, to be sure that the *relA* deletion does not simply decreases transcription of the whole operons or antisense RNAs. Supplemental data 3 and 4 show the list of genes that are differentially expressed depending on the AlpA expression in the wt and *relA* strains – however, since the lists of these genes are different for the two strains, their direct comparison is complicated. In particular, it would be good to compare the numbers (fold-changes in gene expression) for the same list of genes in both strains in parallel, including the first genes before

the terminator. Furthermore, even 30% changes in the level of AlpA expression in the *relA* strain may translate to much larger changes in the expression of its target transcripts, depending on its actual affinity to RNAP and its free concentration in the cell. Importantly, the observed changes in the levels of expression of the target genes are not 1000-fold (this large change is in cell titers, which non-linearly depend on changes in gene expression), but much smaller (only 2-fold for *alpD*, for example, line 346). Furthermore, given the fact that neither deletion of the omega subunit nor deletion of both DksA factors change antitermination by AlpA, the actual mechanism of ppGpp-dependent activation remains obscure, and it may be either direct or indirect. So, I suggest to remove the statement that AlpA is a ppGpp-responsive antiterminator, until the mechanism explaining the observed effects is clear.

Response 4

We have included a Supplementary Table in support of Figs 5b and 5c that shows the requested quantitation from the RNA-Seq studies (*new* Supplementary Table 2 in the Supplementary Information Section). This directly compares the fold effect of ectopic AlpA on the expression of certain sense and antisense transcripts shown in the figure in both WT cells and cells of the $\Delta relA$ mutant. This also includes the effect of AlpA on expression of the gene upstream of each of the terminators indicated in the Figure. In addition, the new Table includes a comparison between the abundance of the indicated transcripts in the absence of AlpA in the WT and $\Delta relA$ mutant backgrounds (i.e. in cells containing the empty control vector). These results, which indicate that there is no general decrease in expression due to the $\Delta relA$ mutation, support the idea that AlpA is more active at these locations in WT cells than in cells of the $\Delta relA$ mutant (by a factor of 2-3) (*new* Supplementary Table 2). Because we would have to identify each gene that precedes a given terminator by manually inspecting the RNA-seq read traces, we have not attempted to generate an additional list in which the fold changes for the WT and $\Delta relA$ mutant strain upon ectopic expression of *alpA* are provided in parallel. Finally, we have removed the statement that AlpA is a ppGpp-responsive antiterminator from the title and abstract and now raise this as a possibility in the Discussion Section.

4) Overexpression of TAP-tagged AlpA reveals its binding to only 6 chromosomal regions (Fig.1); however, in another experiments overexpression of AlpA results in significant changes in the expression of hundreds of genes and antisense transcripts (most of which are activated). Are the levels of AlpA expression comparable in these two cases? Even if activation of transcription occurs through nonspecific binding of AlpA to the elongation complex, is not it expected that AlpA should be enriched in these regions in the ChIP experiments? For the remaining five peaks in the ChIP experiments, can a putative binding site for AlpA be found within the target region? Is transcription of these loci activated by AlpA?

Response 5

We think that the presence of the TAP-tag on AlpA likely reduces its activity; thus, at the concentrations of AlpA-TAP used in the experiments in Fig.1 we see association of AlpA-TAP only with the *PA0807* promoter region and part of the *PA0807-PA0831* locus (the most strongly regulated locus). For the remaining five peaks we cannot identify a putative binding site for AlpA and for four of the corresponding loci, transcription is not activated by AlpA. For the one peak (peak 2 in Supplementary Table 1) where expression of an antisense transcript is activated by AlpA, transcription readthrough of a putative terminator occurs several kb upstream of the peak; therefore, we think it unlikely that the ChIP-seq peak at this location is mechanistically relevant.

5) Deletions of DksA/DksA2 in fact increase the activity of reporter constructs in response to expression of pAlpA (~2-fold) (Fig. S10b). Do the authors have any explanation? So, is it possible to say that DksA counteracts the effects of AlpA on termination, e.g. by promoting

termination?

Response 6

The reviewer raises an interesting possibility here. We don't currently have an explanation for this observation, but in future work we'd like to test whether DksA, which interacts with the secondary channel, might serve to limit the association of AlpA with portions of RNAP that constitute part of the secondary channel.

6) The schemes of DNA constructs containing promoter fragments could be added to Fig. S6, to make it more clear.

Response 7

We now include schemes of the promoter constructs for clarity (*revised* Fig. S6a and b).

Reviewer #2 (Remarks to the Author):

The authors have patiently and thoroughly responded to the 56 comments made by the 4 referees point-by-point. Although the absence of an in vitro system with purified AlpA makes some conclusions speculative, the authors have appropriately toned down these conclusions in response to the reviewers' comments. I have no further comments.

Response 8

We thank the reviewer for their comments.

Reviewer #3 (Remarks to the Author):

All of my comments have been addressed. I think this is a clear and well executed paper and deserves publication.

to note: the colouring and layout of the DNA sequences in Fig 3a has gotten messed up in the merged PDF - something to check

Response 9

We thank the reviewer for their comments.

Reviewer #4 (Remarks to the Author):

The authors have provided a revised and improved manuscript and included additional data to response to previous comments. The revisions provide greater clarity and new evidence to address many comments from myself and the other reviewers. However, I still have several concerns outlined below.

Response 45. I remain concerned about the interpretation of results generated with the ectopic AlpA expression which is ~27X higher than those levels achieved with DNA damage and 270X higher than basal levels according to the authors. The authors confirm that this overexpression is not physiological and that the transcriptional readthrough of many genes likely does not occur in response to DNA damage. In my opinion, there is thus no direct evidence that the bypassing of termination sequences (Fig 5b) occurs under physiological conditions (such as DNA damage), and if it doesn't, what is the biological relevance of the observed results, which only occur as a result of excessively high AlpA expression? This experimental system and the evidence presented thus fail to convince me that those results (Fig 5b) provide meaningful

biological insight.

Response 10

In re-reading the manuscript we realized we had failed to clearly articulate the importance of the findings illustrated in Fig. 5b. We believe they are important for two principal reasons. First, they offer a possible explanation for our surprising finding that high levels of AlpA are toxic even in the absence of *alpBCDE* (Fig. 5a). Second, they could have mechanistic implications for how AlpA functions at physiologically relevant concentrations. In particular, with respect to toxicity, the results in Fig. 5b are presented as evidence that ectopic high-level expression of *alpA* appears to result in readthrough of termination sequences at many genomic regions (and thus explain why AlpA positively regulates the expression of so many sense and anti-sense transcripts under these conditions; Supplementary Data 3 and 4). Such widespread readthrough of transcription terminators could account for AlpA's toxicity under these conditions. Indeed, widespread readthrough of transcription terminators has been suggested to explain the toxicity that results when the transcription termination factor Rho is depleted in *Mycobacterium tuberculosis* (Ref. 38). We have now added text to the Results and Discussion Sections to highlight this important point (lines 300-301 and 433-438). With respect to mechanism, if the widespread readthrough of transcription terminators observed upon ectopic production of AlpA results from AlpA loading onto RNAP in solution, it would suggest that AlpA acts through a mechanism that is distinct from that of λ Q; according to the current model, λ Q would not be expected to be able to load onto RNAP in solution regardless of how high the intracellular concentration was. Our findings with ectopically produced AlpA therefore raise the possibility there could be mechanistic differences between how AlpA and λ Q exert their regulatory effects (as we allude to in the Discussion; lines 429-432). We therefore think it is important to include these data. We note that important insights into the activities of a variety of transcription regulators have been made through studying their regulatory effects at non-physiologically relevant concentrations.

Response 46 The authors have included a time-lapse microscopy which support the observed cell lysis phenotype. With regards to the toxicity of AlpA ectopic expression, the authors state that the low viability encountered with AlpA ectopic production may not be related to cell lysis per se and toxicity could be due to various mechanisms, potentially unrelated to PCD. In the absence of a specific mechanism of toxicity or interaction to explain the modulatory effect of ppGpp, the observation that ppGpp modulates the activity of AlpA is of relatively modest mechanistic significance since ppGpp has such pleiotropic effects.

Response 11

We think our finding that ppGpp may modulate the activity of AlpA could have physiological relevance as *alpA* expression is switched on in response to DNA damage and DNA damage has been shown to result in increased ppGpp levels in the cell, at least in *E. coli*.

Minor comments:

Response 49: The Tn mutagenesis was performed in order to determine the genes involved in the toxic effects of AlpA ectopic expression. It would be informative to explicitly mention that the role of PA4114 was not further investigated.

Response 12

We now explicitly mention that the role of PA4114 was not further investigated (line 325-326).

Response 50: Line 117 The authors state that "These ChIP-Seq analyses revealed that AlpA

associates with 6 regions of the PAO1 chromosome... Fig 1b and Supp table 1". The Fig 1b report on the region PA0807-0830 and the Supp Table 1 is the RNA seq data for *alpA* mutant vs WT with cipro. The authors should report what the other 5 regions are. Since the authors do not think that these 5 other regions are relevant, this should be stated explicitly.

Response 13

These regions are reported in Supplementary Table 1 (not Supplementary Data 1) in the Supplementary Information file. Because Supplementary Table 1 is small the journal requires it to be included in the Supplementary Information section after the Supplementary Figures. (Supplementary Data 1 is the RNA-seq data for the *alpA* mutant vs WT with cipro.)

Response 48 The putative AlpA binding sequence should be explicitly stated in the text or figure legend for clarity, as it is only shown in Fig 3. Furthermore, the author should also report where this binding motif is found in the genome (as per Response 34) and discuss what those results mean since the other genes are not identified in the RNA-seq or ChIP-Seq analyses to be AlpA regulated.

Response 14

We now include the putative AlpA binding sequence in the legend to Fig. 3 as suggested by the reviewer. We don't think that reporting where else in the genome this motif is found is useful to a reader as we don't know whether the putative AlpA binding motif is sufficient for binding or sufficient to confer control by AlpA. Indeed, it may be that additional sequence elements are required for AlpA activity. We note that for λ Q, the Q binding element (QBE) alone does not suffice; a sequence resembling a -10 motif that acts to pause RNA polymerase is also required. This pause-inducing sequence is positioned close to, and downstream of, the transcription start site for PR' and serves to allow λ Q bound at the QBE to stably associate with the transcription machinery. Further studies will be required to determine whether or not there are additional sequence elements that are required for AlpA activity at the target promoters we have identified.